# Dual roles of neutrophils in metastatic colonization are governed by the host NK cell status

Peishan Li[1,2,4], Ming Lu [2,3,4], Jiayuan Shi [2], Li Hua[2], Zheng Gong[2], Qing Li[2], Leonard D. Shultz[2] & Guangwen Ren [2✉]

The role of neutrophils in solid tumor metastasis remains largely controversial. In preclinical models of solid tumors, both pro-metastatic and anti-metastatic effects of neutrophils have been reported. In this study, using mouse models of breast cancer, we demonstrate that the metastasis-modulating effects of neutrophils are dictated by the status of host natural killer (NK) cells. In NK cell-deficient mice, granulocyte colony-stimulating factor-expanded neutrophils show an inhibitory effect on the metastatic colonization of breast tumor cells in the lung. In contrast, in NK cell-competent mice, neutrophils facilitate metastatic colonization in the same tumor models. In an ex vivo neutrophil-NK cell-tumor cell tri-cell co-culture system, neutrophils are shown to potentially suppress the tumoricidal activity of NK cells, while neutrophils themselves are tumoricidal. Intriguingly, these two modulatory effects by neutrophils are both mediated by reactive oxygen species. Collectively, the absence or presence of NK cells, governs the net tumor-modulatory effects of neutrophils.

[1] Key Laboratory of Experimental Teratology, Ministry of Education and Department of Molecular Medicine and Genetics, School of Basic Medical Sciences, Cheeloo College of Medicine, Shandong University, 250012 Jinan, Shandong, China. [2] The Jackson Laboratory, Bar Harbor, ME 04609, USA. [3] Department of General Surgery, Huashan Hospital, Cancer Metastasis Institute, Fudan University, 200040 Shanghai, China. [4] These authors contributed equally: Peishan Li, Ming Lu. ✉email: Gary.Ren@jax.org

Neutropenia, a condition characterized by low counts of host neutrophils, is a common complication in cancer patients receiving cytotoxic therapies such as chemotherapy and radiotherapy[1]. To prevent neutropenia-related infections, cancer patients are usually administered a hematopoietic growth factor-granulocyte colony-stimulating factor (G-CSF), which stimulates the de novo generation of neutrophils through bone marrow (BM) hematopoiesis[2]. However, recent preclinical studies showed that exogenous G-CSF administration and other pathological conditions associated with neutrophil expansion could potentially accelerate solid tumor metastases[3–6]. In cancer patients, increased neutrophil counts (neutrophilia), and a higher neutrophil-to-lymphocyte ratio were also reported to be associated with adverse outcomes including metastatic relapse in multiple types of solid cancers such as breast cancer, gastric cancer, pancreatic cancer and melanoma[7–11].

Increasing evidence reveals that neutrophils exert effects on tumors, particularly metastasis-regulating effects, at the organ colonization step - the most rate-limiting stage of the metastatic process[4,12]. At the metastatic sites, neutrophils function to repress the resident anti-tumor immunity causing an immunosuppressive microenvironment favoring the colonization and outgrowth of the disseminated tumor cells (DTCs)[6,13]. In addition to this immunosuppressive capacity, neutrophils have also been shown to facilitate the DTC extravasation, secrete tumor-trophic factors and awaken the dormant DTCs via formation of neutrophil extracellular traps (NETs)[4,13,14]. Together with other organ resident cells, neutrophils nourish the pre-metastatic and metastatic niche formation[15].

Studies of neutrophil-mediated immunosuppression have mostly focused on inhibition of T cell functions at the primary tumor sites[16,17]. Primary tumor-associated neutrophils, also called myeloid-derived suppressor cells (MDSCs), express high levels of inducible nitric oxide synthase (iNOS) and arginase which serve as the main mechanisms for T cell suppression[18,19]. In the tumor microenvironment, iNOS-expressing neutrophils generate nitric oxide (NO), and a series of NO-derived reactive nitrogen species which induce T cell apoptosis and block T cell activation and effector functions[18,20]. At the same time, tumor-associated neutrophils also express arginase-1 that diminishes CD3ζ expression on T cells thereby impairing T cell functions through reduction of bioavailable L-arginine in the microenvironment[19,21]. In addition to NO and arginase, production of reactive oxygen species (ROS) by tumor-associated neutrophils also serve as a mechanism to suppress the antigen-specific response of CD8[+] T cells[22]. Compared to the extensive studies of neutrophil-mediated immunoregulation in the primary tumor microenvironment, neutrophil modulation of organ-resident anti-tumor immunity at metastatic sites has received less attention and remains poorly defined[16,17]. Limited evidence was obtained from mouse models of breast cancer lung metastasis. In the lung metastatic niche, lung-infiltrating neutrophils have been shown to restrain the anti-tumor CD8[+] T cell response by producing iNOS[6], and were also capable of suppressing intraluminal natural killer (NK) cells, although the underlying mechanisms were not fully characterized[13].

In contrast to the pro-tumoral effects described above, neutrophils also have documented anti-tumoral effects at the primary tumor and metastatic sites in animal models[23–25]. In these animal studies, the anti-tumoral abilities of neutrophils were mainly mediated by their direct cytotoxic effect on tumor cells via neutrophil-derived NO, hydrogen peroxide (H2O2), and the NET structures, as well as anti-proliferative effects through the production of inflammatory cytokines such as IL-1β[24–27]. In humans, the tumoricidal activity of neutrophils has been known for several decades[28,29]. It was reported that human neutrophils isolated from healthy donors are highly toxic to tumor cells, but that this activity was greatly reduced in neutrophils derived from cancer patients[30]. In addition, the tumoricidal activity of human neutrophils appeared to be cancer cell-specific as they were not effective in killing nonmalignant normal cells[30,31].

Therefore, accumulated data from preclinical and clinical studies clearly demonstrate opposite roles of neutrophils in the regulation of tumor growth and metastasis. Given the growing interest in neutrophil targeting as an emerging cancer treatment approach[32], it is critical to determine the basis of the apparent discrepancies about neutrophil interaction with cancer cells. In previous studies, either pro-tumoral or anti-tumoral neutrophil effects were overemphasized under specific in vivo or in vitro circumstances. In this work, we sought to understand the dual tumor-modulatory effects of neutrophils in the same animal models and the same ex vivo culturing conditions. Furthermore, previous studies used a wide variety of different mouse strains, with different genetic backgrounds, for the mouse cancer models. Thus, the genetic variation among mouse strains could also contribute to the contradictory conclusions drawn for neutrophil interactions with tumor cells because immune phenotypes vary greatly among mice with different inbred genetic backgrounds[33,34]. In particular, mice with different levels of immune system integrity, such as immunocompetent versus immunodeficient mice, were variously utilized in cancer studies focusing on neutrophils[3,4,13,24].

Based on the above facts, in the present study we set out to test the hypothesis that the integrity of the host immune system may account for the discrepant roles of neutrophils in mouse cancer models. We perform side-by-side experiments using the same breast tumor models in both immunocompetent and immunodeficient mice. Our results show that neutrophils exert opposite effects in regulation of metastatic colonization in NK cell-competent versus NK cell-deficient mice. Ex vivo mechanistic studies reveal that neutrophils suppress the tumoricidal activity of NK cells, while they themselves are also tumoricidal though to a lesser extent compared to NK cells. Thus, in the absence of NK cells, neutrophils are metastasis-inhibitory, but they show a net effect of metastasis promotion in the presence of NK cells. Our results suggest that the host immune cell status is a predominant factor for neutrophil-mediated modulation of metastasis. It is necessary to evaluate a patient's immune status in clinical management of G-CSF-based neutropenia treatment, as well as in developing new neutrophil-targeting cancer therapeutics.

## Results

**G-CSF injection has distinct metastatic outcomes in mice.** Neutrophils play a critical role in solid cancer metastases but they have been shown to possess both metastasis-promoting and -suppressing capacities. Previous work suggests that neutrophils exert these metastasis-modulating effects as DTCs colonize the metastatic organs[35]. In this study, we first assessed the capability of G-CSF-induced neutrophil expansion (Supplementary Fig. 1a–c) to modulate metastasis using an experimental metastasis model in which metastatic colonization can be quantified (Fig. 1a). Consistent with previous studies[3,13], exogenous G-CSF did augment E0771 breast tumor cell colonization in the lung in immunocompetent host mice (Fig. 1b). To determine whether the immune system integrity affects the role of neutrophils in metastatic colonization, we employed the same experimental metastasis model in immunodeficient host strains including NOD-scid mice, which lack T and B cells, and NOD-scid IL2ry[null] (NSG) mice, which lack T, B and NK cells. Unexpectedly, opposite results were obtained in these two recipient strains (Fig. 1c, d). While G-CSF pre-treatment showed a pro-metastatic effect in NOD-scid recipients similar to

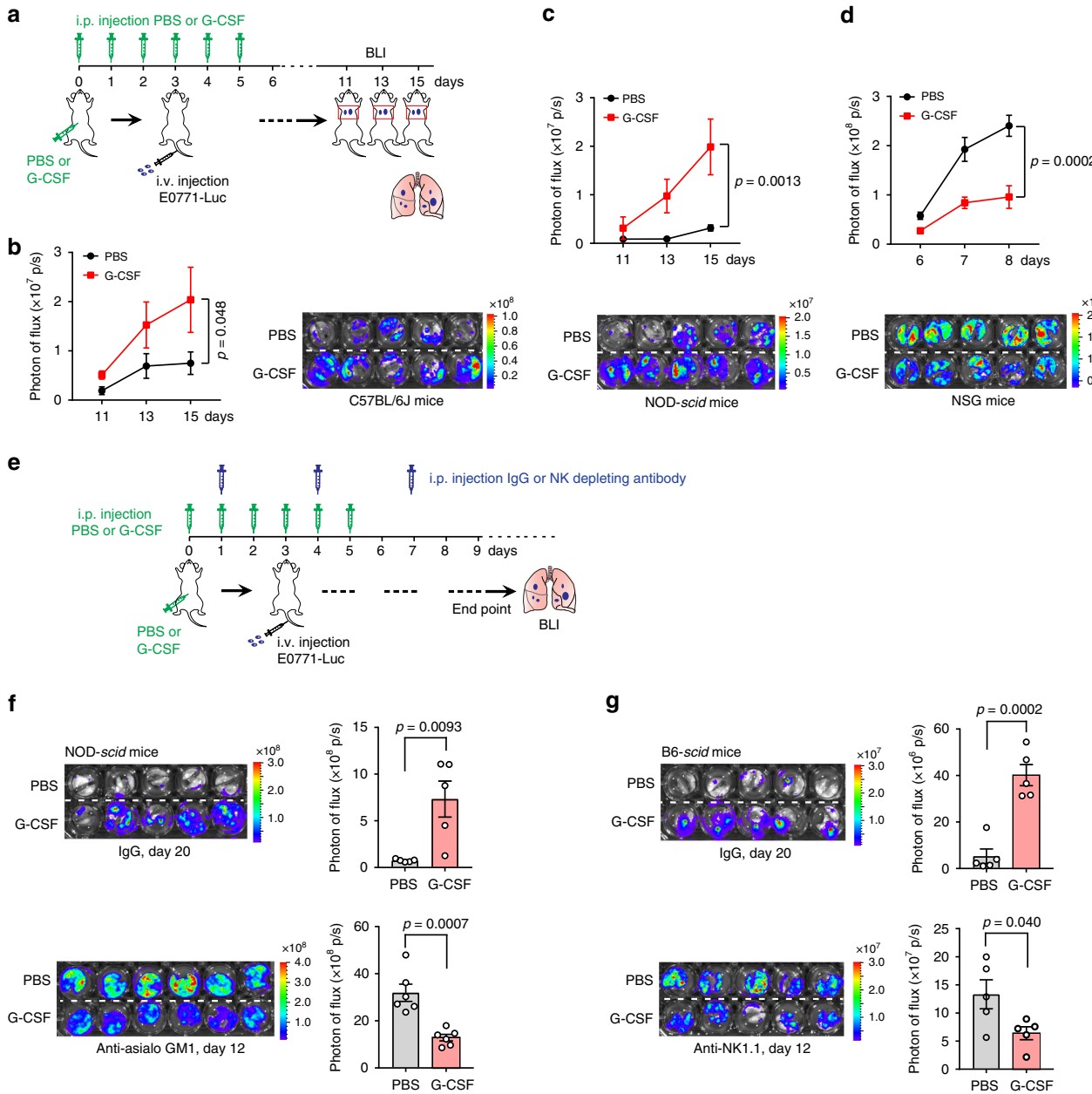

**Fig. 1 G-CSF-induced neutrophilia exerted pro-metastatic and anti-metastatic effects in NK cell-competent and NK cell-deficient mice, respectively. a–d** G-CSF administration showed distinct effects on breast tumor cell colonization in the lungs of C57BL/6J mice (**b**), NOD-*scid* mice (**c**), and NSG mice (**d**). As depicted in **a**, all mice received intraperitoneal (i.p.) injection of recombinant mouse G-CSF (2.5 μg per mouse) for 6 consecutive days. On day 3, the mice were implanted with E0771-Luc cells by intravenous (i.v.) injection. The metastatic progression of E0771-Luc cells was then monitored by bioluminescence imaging (BLI). Quantification of photon flux and comparison of metastatic colonization in the lungs of mice are shown (**b**, left; **c** and **d**, top), and the endpoint bioluminescence images of lungs are shown (**b**, right; **c** and **d**, bottom). $n = 5$ mice per group. *P* values were determined by two-way ANOVA (PBS group versus G-CSF group). **e–g** NK cell depletion converted G-CSF-induced pro-metastatic effect to anti-metastatic. As depicted in **e**, all mice first received i.p. injection of recombinant mouse G-CSF (2.5 μg per mouse) for 6 consecutive days. To deplete NK cells, NOD-*scid* mice (**f**) and B6-*scid* mice (**g**) were i.p. injected with anti-asialo GM1 (25 μl per mouse) and anti-NK1.1 (25 μg per mouse), respectively, every 3 days starting from day 1. On day 3, the mice were implanted with E0771-Luc cells by i.v. injection. At the endpoint, the metastatic progression of E0771-Luc cells in the lungs was detected by ex vivo BLI. The endpoint bioluminescence images (left) and the quantification of photon flux of lungs (right) are shown. $n = 5$ (IgG) or 6 (anti-asialo GM1) NOD-*scid* mice per group (**f**). $n = 5$ B6-*scid* mice per group (**g**). *P* values were determined by unpaired two-tailed *t*-test. Data are represented as mean ± SEM. Source data are provided as a Source data file.

that in immunocompetent C57BL/6J mice (Fig. 1c), it strikingly suppressed the metastatic colonization of E0771 cells in NSG mice (Fig. 1d).

A major immune system difference between NOD-*scid* and NSG hosts is, respectively, the presence and absence of NK cells[36] (Supplementary Fig. 1d). In NOD-*scid* mice, NK cells retained

tumoricidal activity, though to a lesser extent than measured in the immunocompetent C57BL/6J mice (Supplementary Fig. 1e), which was consistent with previous studies[37]. To further exclude the possibility of other cellular differences between the NOD-*scid* and NSG strains besides NK cell presence, we performed NK cell depletion using anti-asialo GM1[38] and anti-NK1.1[39] in NOD-*scid*

and B6-*scid* mice, respectively. As shown in Fig. 1e–g and Supplementary Fig. 1f–g, depletion of NK cells converted the G-CSF-induced pro-metastatic effect to an anti-metastatic effect. We therefore reasoned that the presence or absence of host NK cells has a dominant role in determining which effect is produced by G-CSF-expanded neutrophils in metastatic colonization. In other words, the integrity of the host immune system is a determinant in neutrophil-mediated metastasis regulation.

**Tumor-derived G-CSF is pro- or anti-metastatic in vivo**. Other than exogenous G-CSF administration in treatment of neutropenia, tumor cell-intrinsic G-CSF expression is another clinical condition that causes elevated neutrophil counts[40,41]. The high expression of G-CSF in tumor cells or tissues has been associated with more robust metastatic potential in preclinical tumor models[3,13], and poor prognoses in patients with solid tumors[42,43]. To further evaluate whether the presence or absence of NK cells is a key factor in determining the pro- vs anti-metastatic roles of G-CSF and neutrophils, we constructed *G-csf*-overexpressing E0771 cells (E0771-*g-csf*). As expected, compared to the E0771 parental line, implantation of E0771-*g-csf* cells resulted in a striking increase of host neutrophils but not of other myeloid lineage cells in the lung and peripheral blood (Supplementary Fig. 2a, b).

Next, we adopted a modified experimental metastasis model in which E0771 and E0771-*g-csf* cells were first orthotopically implanted in mice to establish non-inflammatory (neutrophil^low) and inflammatory (neutrophil^high) tumor-bearing host conditions, respectively (Supplementary Fig. 2c). At the pre-metastatic stage in these orthotopic tumor models (Supplementary Fig. 3a), the mice received intravenous injections of luciferase-labeled E0771 cells (E0771-Luc) (Fig. 2a). By monitoring the E0771-Luc cell progression in the lung by bioluminescence imaging (BLI), we were able to compare differences between the neutrophil^high (E0771-*g-csf*) vs non-inflammatory (E0771) host conditions in their accommodation of tumor cells. Consistent with the exogenous G-CSF pre-treatment results, tumor cell-intrinsic *G-csf* overexpression and the resulting host neutrophilia, were again shown to be pro-metastatic in NOD-*scid* mice, but anti-metastatic in NSG mice (Fig. 2b and Supplementary Fig. 4a, b). Similar results were obtained using another genetically distinct AT3 tumor cell line, which was originally derived from a breast tumor arising in MMTV-PyMT transgenic mice[44] (Supplementary Figs. 3b and 4c–e).

Although organ colonization is recognized as a main step for neutrophil-mediated metastasis-regulatory effects[3,4,13], neutrophils have also been reported to be important in other metastatic stages such as primary tumor cell invasion and circulating tumor cell survival[45,46]. To fully understand whether the systemic effect of neutrophils in metastasis also relies on NK cells, we next employed spontaneous metastasis models in NK cell-competent and -deficient mice. To this end, *G-csf*-overexpressed E0771 tumor cells were compared with their parental cells (E0771) for levels of spontaneous metastases in NOD-*scid* and NSG mice. Intriguingly, *G-csf*-overexpression-induced neutrophilia did not affect primary tumor growth, but depending on the host, either enhanced (NOD-*scid*) or mitigated (NSG) the development of spontaneous lung metastases (Fig. 2c–g). Similar results were obtained using the AT3 tumor model (Supplementary Fig. 4f–i). Therefore, overexpression of *G-csf* in tumor cells led to distinct metastatic outcomes in NK cell-competent and –deficient host mice in both experimental and spontaneous metastasis models.

While G-CSF mainly drives neutrophil lineage differentiation, its overexpression in tumor cells also leads to a modest change in other myeloid lineage cells in recipient mice

(Supplementary Fig. 2a, b). To further pinpoint the specific roles of neutrophils in modulation of metastasis, we next selectively depleted neutrophils using anti-Ly6G (clone 1A8) in the modified experimental metastasis model (Fig. 2h). As shown in Fig. 2i, depletion of neutrophils nearly abolished tumor cell *G-csf* overexpression-induced pro- or anti-metastatic effects in vivo. Furthermore, neutrophil depletion was applied in the 4T1 spontaneous lung metastasis model, which is known for strong induction of neutrophilia[3,13]. As expected, anti-Ly6G-mediated neutrophil depletion again showed opposite effects in NK cell-competent and –deficient mice: an anti-metastatic effect in NK cell-competent BALB/c mice and NOD-*scid* mice, but a pro-metastatic effect in NSG mice (Fig. 2j and Supplementary Fig. 5). Interestingly, neutrophil depletion only influenced lung metastasis progression, but did not change primary 4T1 tumor growth in any mouse strains we tested (Supplementary Fig. 5b–d) suggesting neutrophils have a regulatory role specific to metastasis.

Therefore, in both exogenous G-CSF administration and tumor cell *G-csf* overexpression-induced neutrophilia conditions, neutrophilia had an anti-metastatic effect in host mice lacking NK cells but a pro-metastatic effect in host mice with NK cells. These results suggested that the host NK cell status is indeed a master regulator of neutrophils' functions in lung metastasis of breast cancer.

**Neutrophils suppress NK cells but are also tumoricidal**. NK cells and T cells are recognized as the major components of organ resident immunosurveillance during organ-tropic metastasis[47]. Specific for metastatic colonization, our results suggested that NK cells played a dominant role because NK cell-deficient NSG mice allowed a strikingly higher level of tumor cell colonization in the lung, in comparison to NOD-*scid* and immunocompetent recipients (Supplementary Fig. 6). This result is also supported by recent work from other groups showing NK cell depletion was more effective than T cell depletion in restricting lung metastasis in solid cancer models[48,49]. Based on these facts and our in vivo findings as shown in Figs. 1, 2, we propose a tricellular interaction model to explain the observed opposite metastasis-modulating effects of neutrophils in hosts that differ in immune system integrity: both neutrophils and NK cells are tumoricidal, but NK cells possess a more robust cytotoxic capacity than neutrophils. In the absence of NK cells (NSG hosts), neutrophils show a net tumoricidal/anti-metastatic effect while, in the presence of NK cells (NOD-*scid* and immunocompetent hosts), neutrophils suppress the more tumoricidal NK cells and therefore have the net effect of promoting metastatic colonization (Fig. 3a).

Our first examination of the proposed three-way interactions among neutrophils, NK and tumor cells used in vivo monitoring of infused tumor fates during the very early stage of metastatic colonization, a time window when NK cells and neutrophils kill most of the invading tumor cells[12,13]. Consistent with the long-term outcomes shown in Fig. 2a, b, the effects of tumor cell *G-csf*-overexpression-induced neutrophilia were again host dependent: pro-survival in NOD-*scid* (Fig. 3b, c) but anti-survival in NSG mice in vivo (Fig. 3d, e), with the effects evident as early as 2–4 h post-injection. To determine whether the opposing patterns of tumor cell survival were due to differences in tumor cell killing under different host conditions, we next measured the tumoricidal activities of neutrophils and NK cells separately and in combination ex vivo. Based on the immunostaining of neutrophils, NK cells and tumor cells in lung sections from tumor-free and tumor-bearing mice (Supplementary Fig. 7) and the cell-cell

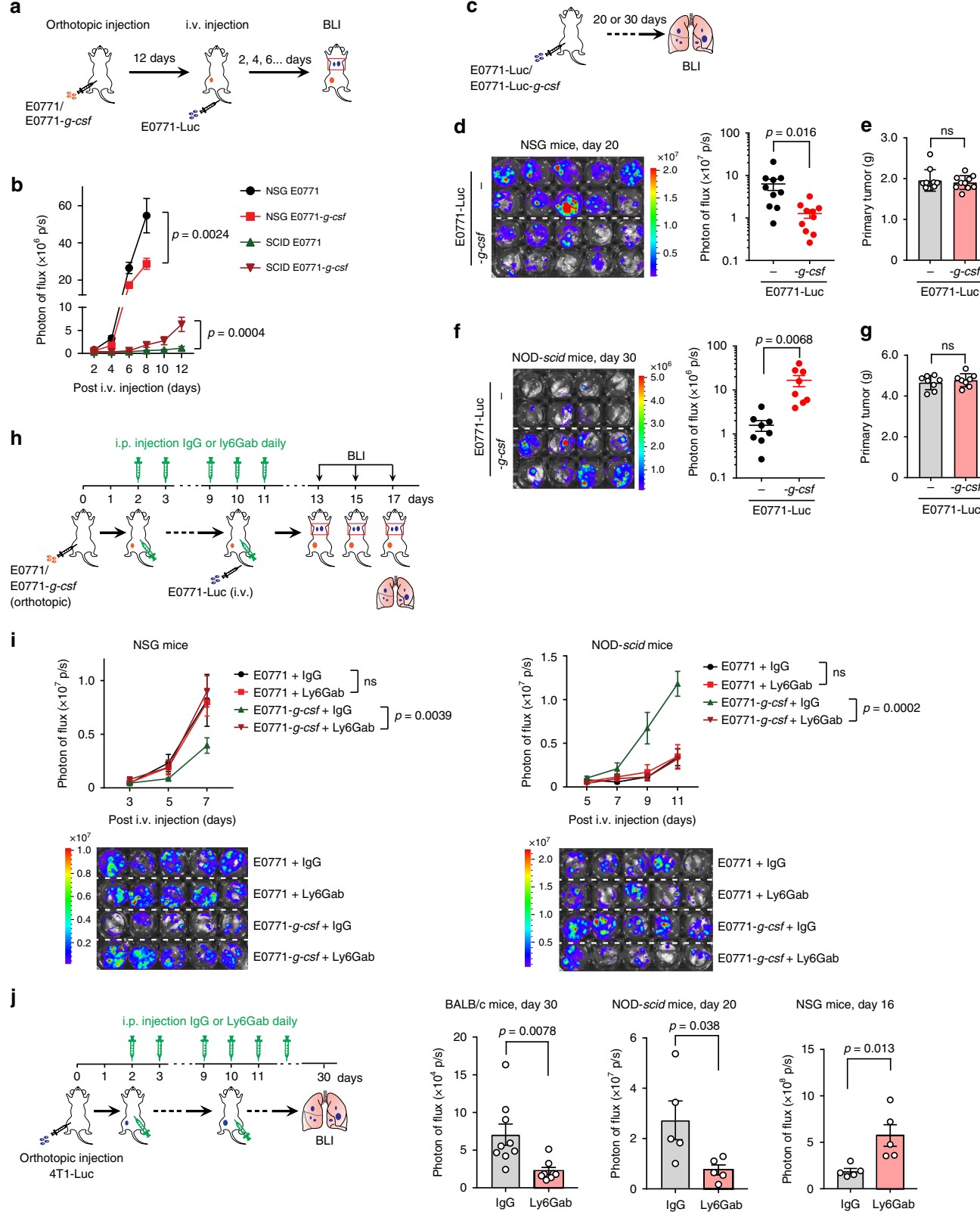

ratios applied in previous reports[24,50], we adopted the ratios of 5–20:1 (neutrophil: tumor cells) and 10–20:1 (NK: tumor cells) to produce the following ex vivo results.

First, lung-infiltrating neutrophils were indeed tumoricidal and, interestingly, the tumor-killing ability of lung neutrophils did not vary based on host origin. Killing ability was similar whether neutrophils were isolated from NK cell-competent or NK cell-deficient mice (Fig. 3f), or from mice with non-inflammatory or inflammatory conditions (Fig. 3g).

Second, lung-infiltrating NK cells were far more effective than neutrophils in killing tumor cells, but their tumoricidal ability was substantially reduced under the neutrophil[high] inflammatory host

**Fig. 2 Tumor cell-derived G-CSF is pro-metastatic or anti-metastatic depending on the host NK cell presence. a, b** Neutrophilia caused by G-csf-overexpression in tumor cells increased metastatic colonization in NOD-*scid* mice, but not in NSG mice in a modified experimental metastasis model. Following the experimental design as in **a**, the metastatic progression of E0771-Luc cells was monitored by BLI. Quantification of photon flux and comparison of metastatic colonization in the lungs of mice are shown (**b**). *n* = 9 mice per group. *P* value was determined by two-way ANOVA. **c–g** As depicted in **c**, NSG mice (**d, e**), and NOD-*scid* mice (**f, g**) were orthotopically implanted with E0771-Luc or E0771-Luc-*g-csf* cells. At the endpoint (day 30 for NOD-*scid* and day 20 for NSG mice), the spontaneous lung metastases were assessed by ex vivo BLI (**d, f**), and the primary tumors were also dissected and weighed (**e, g**). *n* = 10 (**d, e**) or 8 (**f, g**) mice per group. *P* values were determined by unpaired two-tailed *t*-test. ns, not significant. **h, i** Neutrophil depletion abolished the metastasis-regulating effects of tumor cell *G-csf* overexpression in the experimental metastasis model. A schematic diagram was depicted in **h** showing the experimental design. Quantification of photon flux and comparison of metastatic colonization in the mouse lungs are shown on the top, and the endpoint bioluminescence images of lungs are shown on the bottom (**i**). *n* = 5 mice per group. *P* values were determined by two-way ANOVA. ns not significant. Ly6Gab: Anti-Ly6G antibody. **j** In the 4T1 spontaneous metastasis model neutrophil depletion showed anti-metastatic and pro-metastatic effects in NK-cell competent (BALB/c and NOD-*scid*) and NK cell-deficient (NSG) mice, respectively. The lung metastases were examined by ex vivo BLI and quantification of photon flux of lungs are shown. *n* = 9 (IgG group) and 8 (Ly6Gab group) for BALB/c mice; and *n* = 5 per group for NOD-*scid* and NSG mice. *P* values were determined by unpaired two-tailed *t*-test. Data are represented as mean ± SEM. Source data are provided as a Source data file.

condition (Fig. 3g–i and Supplementary Fig. 8a, b). This indicated the possibility for neutrophil-mediated suppression of NK cell functions. Indeed, in both E0771 and AT3 models a significant reduction of the effector IFNγ$^+$ and CD107$^+$ lung-infiltrating NK cells was detected in neutrophil$^{high}$ host mice, in spite of no change in total NK cell number (Fig. 4a–d and Supplementary Fig. 8c–f). This lung NK cell suppression was largely reversed when neutrophils were systemically depleted, a finding that further emphasizes the specific role of neutrophils in modulating the host anti-tumor immunity (Fig. 4e–h). Interestingly, NK cells were not simultaneously affected by neutrophils at the primary tumor sites which again suggests neutrophil-mediated immunosuppression is tissue specific (Supplementary Fig. 8g–i).

Third, to further clarify the triangular relationship among neutrophils, NK cells and tumor cells, we performed a three cell population ex vivo co-culture using lung neutrophils derived from tumor-bearing mice and normal NK cells isolated from naïve mice. Consistent with the results above, either neutrophils or NK cells alone were tumoricidal although neutrophils killed tumor cells less robustly than NK cells (Fig. 4i–k). However, a combination of both neutrophils and NK cells did not increase the tumoricidal effect (Fig. 4i–k) which supports the involvement of neutrophil-mediated NK cell suppression as indicated above. Further, the tri-cellular interaction was independent of the neutrophil source as neutrophils isolated from non-inflammatory and inflammatory host mice showed comparable effects on NK cells and tumor cells (Supplementary Fig. 8j).

Taken together, these in vivo and ex vivo results unambiguously support the proposed tricellular interaction model. On the one hand, both neutrophils and NK cells are tumoricidal, though the latter is more effective, but on the other hand, neutrophils can also suppress the tumorcidal activity of NK cells. Thus, under a given host condition, tumor progression ultimately depends, not on a single factor, but on the interactions among neutrophils, NK cells, tumor cells and quite possibly other components in the tissue microenvironment (Fig. 3a).

**Neutrophils regulate both NK and tumor cells via ROS.** In previous studies, investigation of neutrophil-mediated immunosuppression was primarily focused on T cells which are repressed by a variety of neutrophil-derived immunoregulatory factors such as NO and ROS[6,18,20,23]. In contrast, fewer studies focused on NK cells, the essential anti-tumor host immune cells in the organ metastatic niche. Limited evidence supports the idea that organ-infiltrating neutrophils suppress the ability of NK cells to undergo functional activation[13], but a mechanistic understanding of neutrophil-mediated NK cell inhibition is not yet available.

We set out to determine the molecular basis by which neutrophils suppress NK cell functions by focusing on known immunoregulatory factors including ROS and NO, both of which are recognized to be critical in inflammation and cancer[51–53]. The main bioactive ROS molecules are produced through a series of enzyme-catalyzed reactions: oxygen ($O_2$)→superoxide ($O_2^{\bullet-}$)→hydrogen peroxide ($H_2O_2$)→$H_2O_2$ derivatives (such as hydroxyl radicals and hypochlorous acid)[54]. Among the key enzymes catalyzing these reactions, nicotinamide adenine dinucleotide phosphate (NADPH) oxidase catalyzes $O_2$ to $O_2^{\bullet-}$, whereas catalase converts $H_2O_2$ into $O_2$ and water. NO, another type of redox molecule, can be coupled with the ROS pathway by reacting with superoxide to produce reactive nitrogen species such as peroxynitrite ($ONOO^-$)[54].

In the typical bicellular neutrophil-NK cell co-culture system, selective inhibition of NADPH oxidase by histamine dihydrochloride (HDC) or apocynin, or stimulation of $H_2O_2$ decomposition by catalase remarkably increased the proportion of effector IFNγ$^+$ NK cells which were repressed by lung neutrophils, whereas inhibition of NO was ineffective (Fig. 5a, b). Functionally, blocking the ROS pathway, but not NO, largely abolished lung neutrophil-mediated suppression on the tumoricidal activity of NK cells as determined in the tricellular co-culture system (Fig. 5c). Therefore, the ROS pathway is revealed as a primary mechanism by which lung neutrophils can suppress NK cell functions.

To further delineate the association of neutrophils with the ROS pathway in vivo, we measured endogenous lung ROS levels in mice bearing E0771 and E0771-*g-csf* tumors. The ROS levels in the lung were significantly elevated when the hosts underwent neutrophilia driven by tumor cell *G-csf* overexpression. Such an elevation was markedly abrogated by neutrophil depletion with anti-Ly6G (Fig. 5d) indicating that the burst of ROS in the lungs is primarily, if not exclusively, caused by neutrophils.

As ROS are bioactive species that nonspecifically target diverse types of cells and tissues in inflammation and cancer, the greatly heightened lung endogenous ROS levels prompted us to speculate whether these neutrophil-derived ROS may also influence tumor cells. Using the ex vivo neutrophil-tumor cell co-culture, we observed reductions in neutrophil tumoricidal activity in response to either inhibition of NADPH oxidase by apocynin or HDC, or stimulation of $H_2O_2$ degradation of by catalase (Fig. 5e, f). Therefore, the ROS pathway emerges as a main mechanism which drives neutrophils to nonspecifically target both NK cells and tumor cells at the metastatic site.

In addition to mediating cytotoxic effects from the effector to target cells, ROS have also been reported to participate in modulating intracellular signaling cascades in various types of cells such as neutrophils[55]. To affirm ROS as key neutrophil-

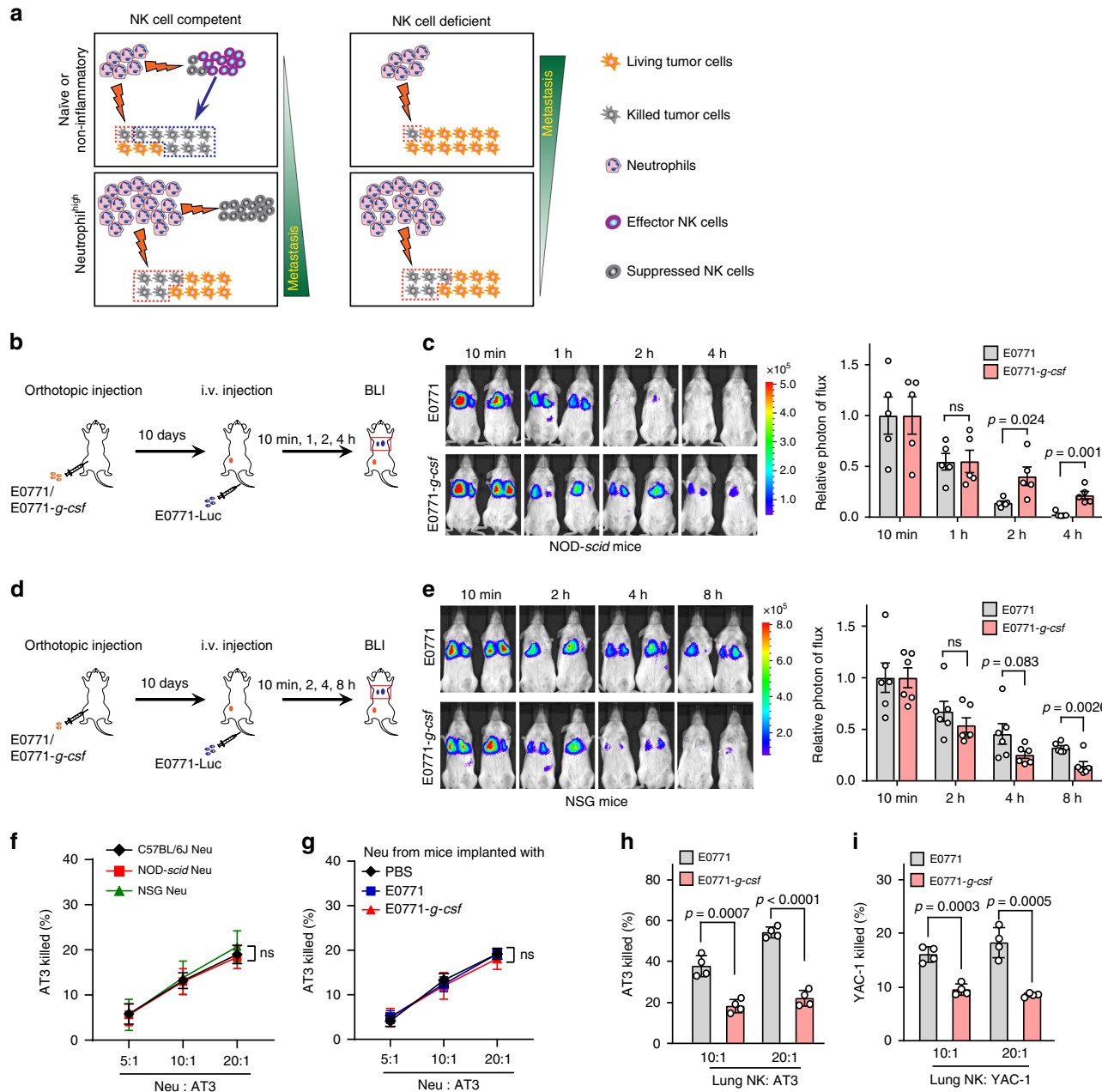

**Fig. 3 The metastasis-modulating effects by neutrophils occur at the early stage of metastatic colonization. a** Model depicting the dual roles of neutrophils in metastatic colonization. In the absence of NK cells, neutrophils exclusively exert their tumoricidal effect thereby showing a net anti-metastatic effect; in the presence of NK cells, neutrophils are also tumoricidal but they suppress the more tumoricidal NK cells leading to a net pro-metastatic effect. **b**–**e** Determination of neutrophilia-induced metastasis regulation at the early stage of colonization. The diagrams in **b**, **d** show the experimental designs in **c**, **e**, respectively. The representative BLI images (left), and quantification of photon flux (right) in each group at various time points are shown in **c**, **e**. $n = 5$ NOD-*scid* mice in **c** and 6 NSG mice in **e** per group. *P* values were determined by unpaired two-tailed *t*-test. **f** Comparison of the tumoricidal activities of neutrophils isolated from C57BL/6J, NOD-*scid* and NSG mice. Lung-infiltrating neutrophils were freshly isolated from E0771-*g-csf* tumor-bearing mice for ex vivo cytotoxic assay against AT3 cells. Statistical significance was determined by two-way ANOVA. Neu: neutrophils. **g**–**i** The tumoricidal capacities of neutrophils and NK cells under non-inflammatory and neutrophil^high inflammatory conditions. E0771 cells and E0771-*g-csf* cells were orthotopically implanted in NOD-*scid* mice to induce non-inflammatory and inflammatory tumor-bearing host conditions, respectively, with PBS injection as a control. The mice were sacrificed at the pre-metastatic stage (day 10), and lung-infiltrating neutrophils were isolated for the ex vivo cytotoxic assay against AT3 cells (**g**). Statistical significance was determined by two-way ANOVA. Lung-infiltrating NK cells were also isolated from the same mice for the ex vivo cytotoxicity assay against AT3 (**h**) or YAC-1 cells (**i**). *P* values were determined by unpaired two-tailed *t*-test (**h**, **i**). Data are represented as mean ± SEM (**c**, **e**) or mean ± SD of four biologically independent cell cultures (**f**–**i**). ns not significant. Source data are provided as a Source data file.

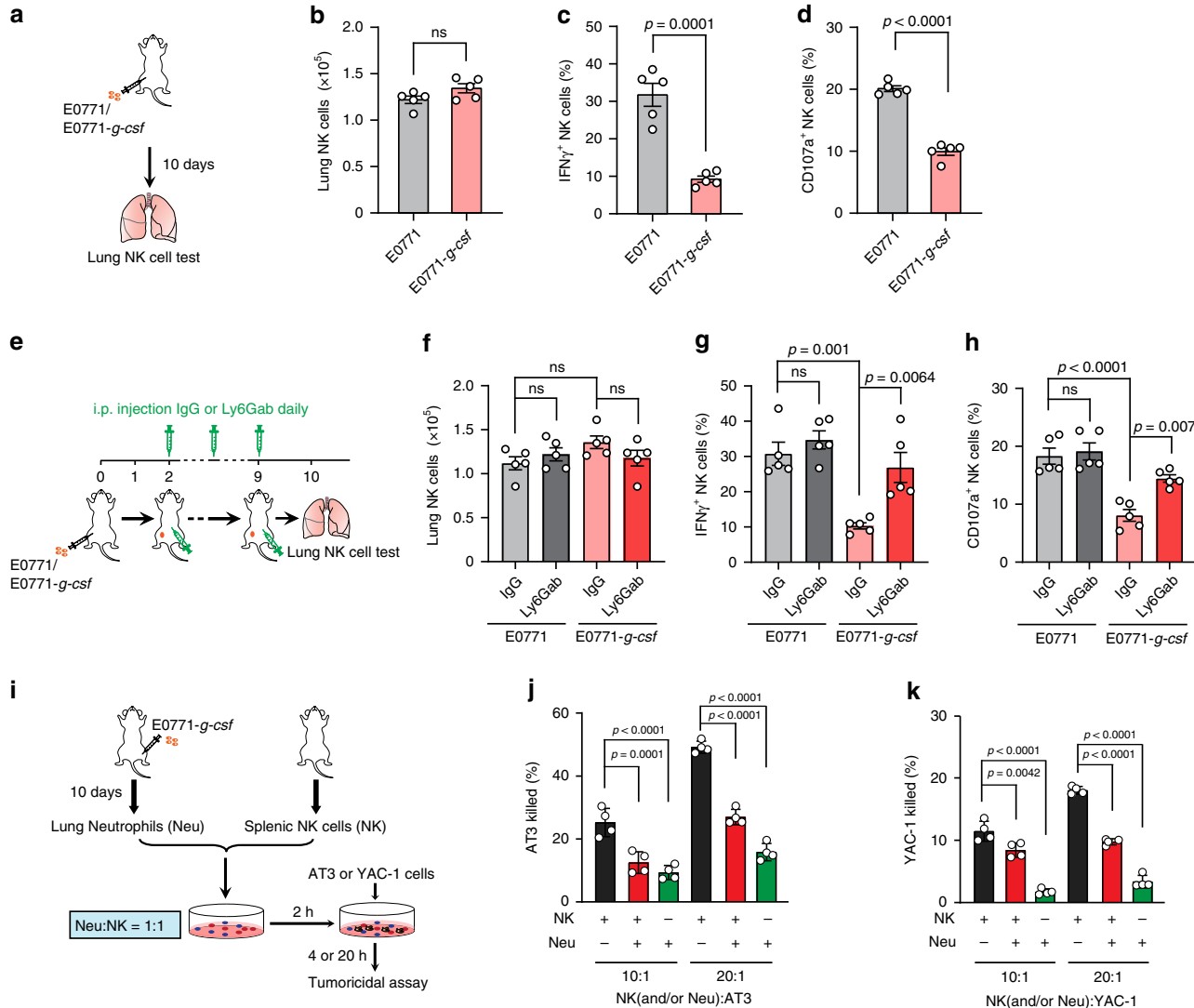

**Fig. 4 Neutrophils inhibit the tumoricidal activity of NK cells. a–d** Neutrophilia induced effector NK cell suppression in the pre-metastatic lungs. As depicted in **a**, NOD-*scid* mice were orthotopically injected with E0771 or E0771-*g-csf* cells. The mice were sacrificed at day 10. The total numbers of lung-infiltrating NK cells were calculated upon analysis by flow cytometry (**b**), and the percentages of IFNγ+ (**c**), and CD107a+ effector NK cells (**d**) were determined by flow cytometry. *n* = 5 mice per group. *P* values were determined by unpaired two-tailed *t*-test. **e–h** Depletion of neutrophils abolished tumor cell *G-csf*-overexpression-induced NK cell suppression. As depicted in (**e**), E0771 or E0771-*g-csf* cells were orthotopically implanted in NOD-*scid* mice, and the mice received control IgG or anti-Ly6G (Ly6Gab) daily 2 days after tumor cell implantation. The mice were sacrificed at day 10, and the total number of lung-infiltrating NK cells were calculated upon analysis by flow cytometry (**f**). The percentages of IFNγ+ (**g**) or CD107a+ effector NK cells (**h**) were determined by flow cytometry. *n* = 5 mice per group. *P* values were determined by one-way ANOVA with Tukey's multiple comparisons test. **i–k** In **i**, a schematic diagram is to show the experimental design in **j**, **k**. Neutrophils were freshly isolated from the lungs of E0771-*g-csf* orthotopic tumor-bearing NOD-*scid* mice, and NK cells were derived from spleens of naïve NOD-*scid* mice. An ex vivo cytotoxicity assay was performed in a co-culture system including neutrophils, NK cells and tumor cells (3 × 10^4 AT3 cells or 1 × 10^4 YAC-1 cells), with the ratios of neutrophil: NK cell: tumor cells at 10:10:1 or 20:20:1. The percentages of AT3 cells (**j**) and YAC-1 cells (**k**) killed by neutrophils and NK cells individually or together were determined. *P* value was determined by one-way ANOVA with Tukey's multiple comparisons test. Data are represented as mean ± SEM (**b–d**, **f–h**) or mean ± SD of four biologically independent cell cultures (**j**, **k**). ns not significant. Source data are provided as a Source data file.

derived extracellular mediators that act on tumor cells and NK cells, we prepared cell-free conditioned medium (CM) from lung neutrophils and tested whether the medium functioned similarly to neutrophil cells towards tumor cells and NK cells. As shown in Supplementary Fig. 9a–c, the lung neutrophil-derived CM was indeed effective in killing tumor cells and suppressing the tumoricidal activity of NK cells. As expected, stimulation of $H_2O_2$ decomposition by catalase, but not inhibition of the intracellular NADPH oxidase, reversed the effects of the neutrophil CM. These results indicated that neutrophil-derived ROS are able to exert extracellular effects on tumor cells and NK cells although the

ROS-mediated intracellular functions in neutrophils are not completely excluded.

**ROS blockade abolishes the dual roles of neutrophils in vivo.** As evidenced by our ex vivo work described above, neutrophils require the ROS pathway to exert both tumoricidical and NK cell suppressive effects. We then wondered whether pharmacological inhibition of the ROS pathway could serve as an effective approach to manage the modulatory functions of neutrophils in metastatic colonization. To this end, we first examined whether administration of an ROS inhibitor would reduce endogenous

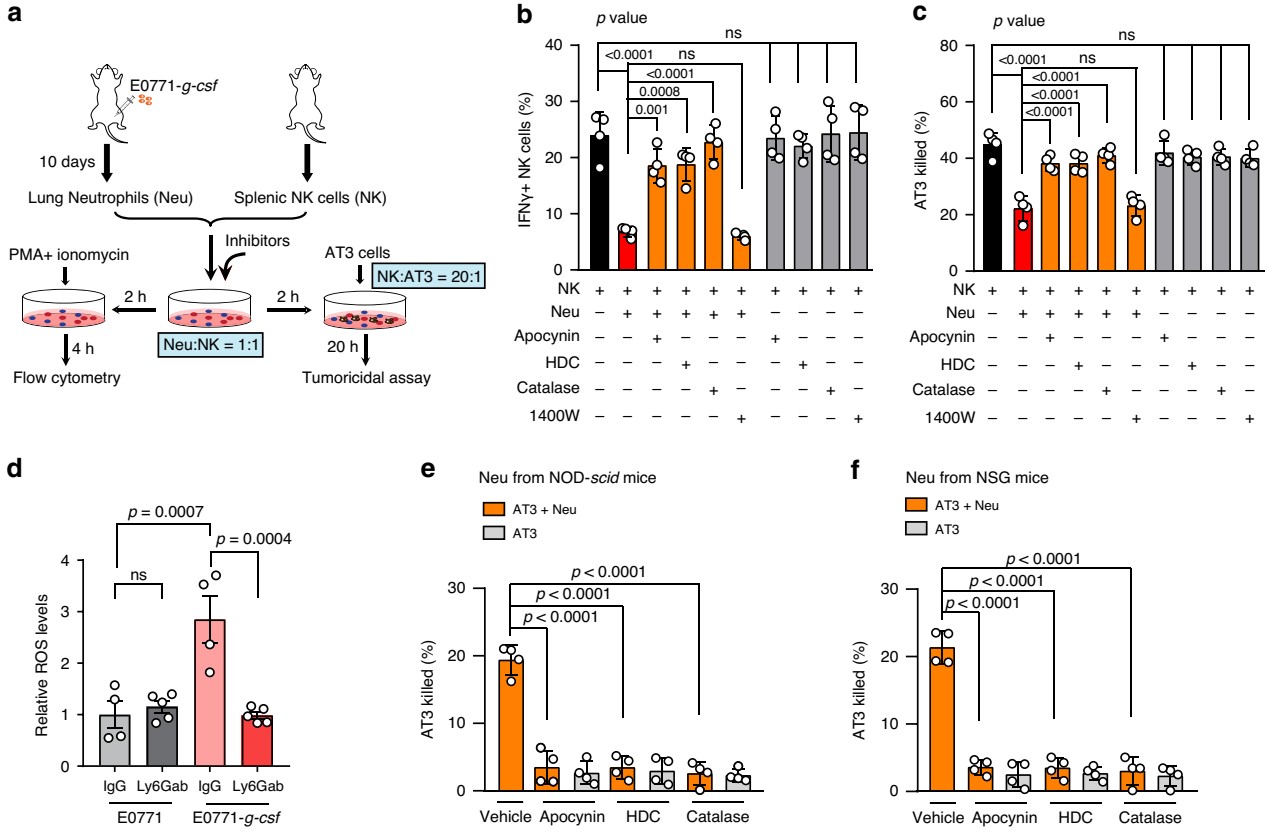

**Fig. 5 The tumoricidal and NK-suppressive effects of neutrophils are both mediated by ROS. a–c** Neutrophils suppress NK cell functions via ROS. As depicted in **a**, in the co-culture system including neutrophils and NK cells, apocynin, histamine dihydrochloride (HDC), catalase or 1400 W was added at the beginning of the co-culture, and the cells were then primed with PMA and ionomycin for 4 hours before determination of IFNγ+ NK cells by flow cytometry (**b**) or for the ex vivo cytotoxicity assay against AT3 cells (**c**). Neutrophils were freshly isolated from the lungs of E0771-*g-csf* orthotopic tumor-bearing NOD-*scid* mice, and NK cells were derived from spleens of naïve NOD-*scid* mice. **d** Depletion of neutrophils reduced the elevated lung ROS levels caused by tumor cell *G-csf* overexpression. NOD-*scid* mice received control IgG or anti-Ly6G antibody (Ly6Gab) daily 2 days after E0771 or E0771-*g-csf* orthotopic implantation, for 10 consecutive days. Then the mouse lungs were isolated for determination of the ROS levels by bioluminescence (probe L-012). The ROS levels were normalized to the average of the values from the E0771 implanted group treated with IgG. $n = 4$ (IgG group) and 5 (Ly6Gab group) mice per group. **e**, **f** The tumoricidal effect of neutrophils was dependent on ROS. Lung-infiltrating neutrophils were freshly isolated from E0771-*g-csf* orthotopic tumor-bearing NOD-*scid* mice (**e**) or NSG mice (**f**) for measuring of their cytotoxic effect on AT3 cells ($3 \times 10^4$ AT3 cells) at the effector: target ratio of 20: 1. Apocynin, HDC or catalase was added at the beginning of culture. Data are represented as mean ± SD of four biologically independent cell cultures (**b**, **c**, **e**, **f**) or mean ± SEM (**d**). *P* values were determined by one-way ANOVA with Tukey's multiple comparisons test. ns, not significant. Source data are provided as a Source data file.

ROS levels in the lung in vivo. Indeed, treatment with the NADPH oxidase inhibitor HDC significantly decreased the lung ROS levels in mice with neutrophilia induced by tumor cell *G-csf* overexpression (Fig. 6a, b). In accordance with the reduced ROS levels, neutrophilia-induced effector NK cell suppression was largely alleviated by HDC treatment, while the total number of lung-infiltrating NK cells remained unchanged (Fig. 6c–e).

We further adopted the modified experimental metastasis model, as outlined in Fig. 2a, to evaluate how blocking neutrophil-derived ROS would ultimately alter the progression of metastasis in NK cell-competent and –deficient recipient mice. The HDC treatment was started at the pre-metastatic stage, when the ROS burst had already emerged (Fig. 6b), and went on into the metastatic stage (Fig. 6f–i). Similar to the effects caused by neutrophil depletion (Fig. 2h–i), ROS inhibition profoundly abrogated the neutrophilia-induced pro-metastatic effect in NK cell-competent NOD-*scid* mice (Fig. 6f, g), and anti-metastatic effect in NK cell-deficient NSG mice (Fig. 6h, i). Thus, these findings indicate that the ROS pathway is essential in dictating the dual roles played by neutrophils in metastatic colonization.

## Discussion

Using mice with different levels of immune system integrity, our in vivo data suggested that G-CSF-expanded neutrophils were anti-metastatic in NK cell-deficient mice, but pro-metastatic in NK cell-competent mice. Further mechanistic studies indicated an interactive relationship among neutrophils, NK cells and tumor cells. Specifically, NK cells are more tumoricidal than neutrophils, and neutrophils exert inhibitory effects on NK cells and cytotoxic effects on tumor cells. Therefore, in the absence of NK cells, neutrophils only showed their tumoricidal ability or a net anti-metastasis effect, while in the presence of NK cells, although neutrophils were still tumoricidal they suppressed the more tumoricidal NK cells resulting in a net pro-metastatic effect (Fig. 3a). Finally, the ROS pathway was revealed to be necessary for neutrophils to exert the dual roles. Taken together our results show that the host immune status, especially the NK cell state, dictates the net contribution of neutrophils to metastasis. Our findings therefore offer a model that provides a coherent interpretation of controversial findings surrounding the role of neutrophils in metastatic colonization.

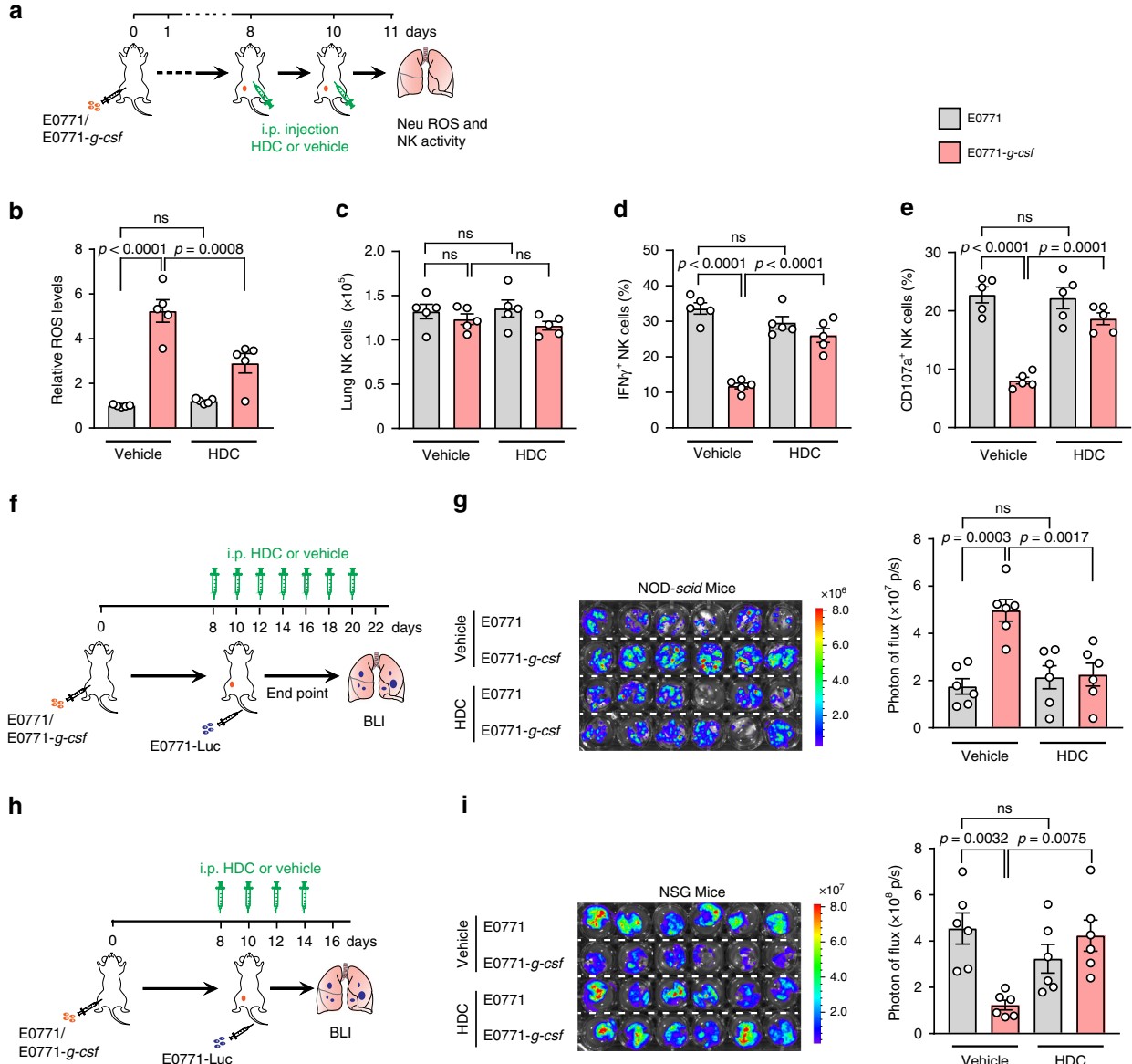

**Fig. 6 Inhibition of ROS pathway abrogates the pro-metastatic and anti-metastatic effects of neutrophils in vivo. a–e** ROS blockade reversed neutrophilia-induced NK cell suppression. As depicted in **a**, ROS inhibitor HDC or the vehicle control was given to the NOD-*scid* mice bearing E0771 or E0771-*g-csf* orthotopic tumors at the pre-metastatic stage (day 8 and 10). On day 11, the mouse lungs were isolated for ROS level determination by bioluminescence (probe L-012) (**b**), total lung-infiltrating NK cell counting (**c**), and measurement of IFNγ+ NK cells (**d**) and CD107a+ NK cells (**e**) by flow cytometry. $n = 5$ mice per group. **f–i** Administration of the ROS inhibitor HDC abolished the tumor cell *G-csf* overexpression-induced metastases-regulatory effects. As depicted in **f**, **h**, NOD-*scid* (**g**) or NSG (**i**) mice were first orthotopically injected with unlabeled E0771 and E0771-*g-csf* cells to generate the non-inflammatory and inflammatory host conditions, respectively. E0771-Luc cells were i.v. infused at the pre-metastatic stage (day 10). From day 8, the mice also received HDC (1mg per mouse) every other day. At the endpoint, the metastatic progression of E0771-Luc cells in the lungs was detected by ex vivo BLI. The bioluminescence images (left) and the quantification of photon flux of lungs (right) are shown (**g**, **i**). $n = 6$ mice per group. Data are represented as mean ± SEM. *P* values were determined by one-way ANOVA with Tukey's multiple comparisons test. ns, not significant. Source data are provided as a Source data file.

Our work also has clinical implications. G-CSF has been widely used in cancer patients who receive cytotoxic therapies and consequently develop neutropenia[2]. The main function of G-CSF is to stimulate the de novo generation of neutrophils through hematopoiesis, although this cytokine has also been suggested to influence other myeloid lineage cell development[56] and to directly act on NK cells[57]. As shown in Supplementary Figs. 1a–c and 2a, b, both exogenous G-CSF administration and *G-csf* over-expression in tumor cells caused a remarkable expansion of neutrophils, with only minimal or insignificant changes in other myeloid lineage cells and adaptive immune cells in host mice.

Furthermore, the direct effect of G-CSF on NK cells was also examined. In this case, exogenous addition of recombinant G-CSF did not change the proportion of effector NK cells or their tumoricidal ability (Supplementary Fig. 10a–c). These data highlighted the specificity of G-CSF in driving host neutrophilia and subsequently regulating metastasis.

The results in Fig. 1 indicated that exogenous G-CSF adminis-tration enhanced tumor cell metastatic colonization in host mice with intact NK cells, but an opposite anti-metastatic effect in host mice lacking functional NK cells. Cancer patients with cytotoxic treatment-induced neutropenia also commonly experience a

collapse of other immune cell populations, including NK cells, which are also susceptible to cytotoxic drugs[58]. Thus, to some extent, these patients resemble NSG mice with their severe immune defects. As such, it is anticipated that exogenous G-CSF administration may lead to a net anti-metastatic outcome in neutropenia patients. Indeed, clinical data reveal that G-CSF treatment was mostly safe and significantly reduced the risk of infection-related mortality and early mortality from all causes in neutropenia patients[2]. Based on our results in mice, patients with high levels of immune defects, such as those with risk of febrile neutropenia[59], may gain more benefit from the G-CSF treatment. In contrast, individuals retaining a relatively intact immune system, particularly functional NK cells, could risk metastatic relapse in response to exogenous G-CSF treatment.

An increase in the neutrophil-to-lymphocyte ratio has been repeatedly reported to be correlated with poor prognoses including distant metastases in various types of solid cancers[8,9]. According to our data, in hosts with neutrophilia the immunosuppressive effect of neutrophils may prevail over their tumoricidal activity and lead to a net pro-tumoral effect. This could happen at the early stages of cancer progression when the host immune system was still in a relatively intact state. In the last decade, strategies targeting hematopoietic growth factors and neutrophil trafficking have been explored in clinical trials to treat advanced solid cancers[60,61]. Our results would suggest that future clinical application would benefit from careful evaluation of each individuals' immune status prior to neutrophil-targeting therapy. In patients with defective NK cells caused by cytotoxic therapeutics or other pathological conditions, the G-CSF- or neutrophil-targeting therapy may abrogate the neutrophils' tumoricidal ability leading to deleterious outcomes. Hence, a personalized immune status assessment should be considered for both clinical use of G-CSF in neutropenia treatment, and as a design component of new neutrophil-targeting approaches to treat solid cancer metastasis.

In addition to the important clinical implications described above, our results also shed light on the ongoing need for optimal selection of preclinical animal models. Our work clearly shows that opposite conclusions could be obtained with regard to the functional contribution of neutrophils in lung metastasis depending on whether data were derived from NOD-scid or NSG mice. Therefore researchers need to be cautious when using these mice, or others that differ in immune system integrity, to study the functions of immune cells in cancer. Owing to the close interactions among different populations of immune cells in the tumor microenvironment, the function of one type of immune cell could be altered by the loss of another type of immune cells[62]. For example, in mouse models of fibrosarcoma and lung carcinoma, depletion of NK cells strengthened the tumor-promoting effect of primary tumor-associated neutrophils via the increased expression of angiogenetic factors in neutrophils[63]. In our study, we also conducted a comprehensive comparison of lung-infiltrating neutrophils isolated from NK cell-competent (C57BL/6J and NOD-scid) and NK cell-deficient mice (NSG). It was shown that lung-infiltrating neutrophils did not significantly differ in host NK cell status, either phenotypically or functionally, including their tumoricidal effect (Fig. 3f), NK cell suppressive effect (Supplementary Fig. 11a), ROS producing levels (Supplementary Fig. 11b) and expression of the key genes relevant to neutrophil biology in cancer (Supplementary Fig. 11c).

Besides differences in immune system integrity, additional immune system variations related to genetic background also need to be considered in murine cancer models. The mouse strain differences in T cell- and NK cell-mediated immunity have been long recognized[64–66], and correspond well to the genetic variation-associated disparity in human immune responses[67].

Therefore, it is plausible to speculate that a large part of the controversial arguments surrounding functions of certain immune cells in cancer could arise because of host mice that differ in immune system integrity, or genetic background.

Our work pinpointed ROS as an indispensable mechanism by which neutrophils exert their dual roles in regulating both NK cells and tumor cells. ROS are signaling molecules that play a central role in host defense against invading pathogens and the progression of inflammatory responses[54]. At infection or wound sites, neutrophils are a key cell population producing a large amount of ROS that not only function to eliminate the pathogens, but also lead to endothelial dysfunction and tissue injury[54]. Thus, ROS is a double-edged sword to the host in infectious and inflammatory diseases. Similarly, the pro-tumoral and anti-tumoral effects of ROS have also been repeatedly revealed in cancer[51]. At low concentration levels, ROS can serve as second messengers for a variety of signaling pathways such as phosphoinositide-3-kinase/Akt cascade, the mitogen-activated protein kinase/Erk cascade, and the nuclear factor κ-B signaling pathway, all of which are critical for tumor cell survival, proliferation, differentiation and metabolism[51]. On the other hand, high concentrations of ROS are toxic to both tumor cells and the host cells, especially the anti-tumor immune cells, leading to tumoricidal and immunosuppressive effects[22,24]. It is therefore not surprising that ROS blockade showed both inhibitory and stimulatory effects on cancer progression in mouse models[68–70]. Our study indicated that neutrophil-derived ROS were able to act on both host NK cells and tumor cells ex vivo (Fig. 5a–f), and consequently showed both pro-metastatic and anti-metastatic effects in vivo (Fig. 6f–i). The host immune system integrity, particularly the NK cell status, was therefore critical to the ultimate function of ROS in determining tumor-regulation in mice. Again, taken together our results suggest a systemic evaluation of a patient's immune status will allow for a more precise and efficacious use of ROS-targeting therapeutics in treating cancer. In our future work, it will also be a fascinating and important research direction to further investigate the influence of neutrophil-derived ROS on distinct lung resident cell subsets during breast cancer progression, as well as in other inflammatory diseases. The results will help us to develop a deeper and more precise understanding of the lung biology in steady and pathological conditions.

## Methods

**Mice.** C57BL/6J (JAX stock #000664), BALB/c (JAX stock #000651), NOD.Cg-Prkdc[scid]/J (NOD-scid, JAX stock #001303), B6.Cg-Prkdc[scid]/SzJ (B6-scid, JAX stock #001913) and NOD.Cg-Prkdc[scid]Il2rg[tm1Wjl]/SzJ (NSG, JAX stock #005557) mice were obtained from the Jackson Laboratory. The mice were fed on a chow diet ad libitum and housed in a specific pathogen-free facility in plastic cages at 22 °C and 40–50% humidity, with a daylight cycle from 6 a.m. to 6 p.m. The animal protocols for the experiments described in this manuscript were approved by the Institutional Animal Care and Use Committee of The Jackson Laboratory. The mice were matched for age and sex in each experiment.

**Tumor models.** For orthotopic tumor transplantation models, tumor cells ($5 \times 10^5$ cells) were re-suspended in 50 μl growth factor-reduced Matrigel (Corning) and transplanted within the fourth mammary fat pads on the left flank of the recipient mice. For experimental metastasis models, tumor cells ($5 \times 10^5$ cells per injection) were re-suspended in 100 μl PBS and infused into the recipient mice by tail vein injection.

**Cell culture.** AT3 and 4T1 cells were maintained in DMEM supplemented with 10% fetal bovine serum (FBS). E0771 and YAC-1 cells were maintained in RPMI 1640 medium supplemented with 10% FBS. All cells were cultured in a 5% CO$_2$ humidified incubator at 37 °C.

E0771-Luc, AT3-Luc and 4T1-Luc cells were derived from their parental cells after infection with luciferase-expressed lentivirus vectors (addgene #17477). Construct-positive cells were selected with puromycin. To overexpress G-csf, E0771 or AT3 cells were infected with the lentiviral vectors expressing the G-CSF protein

coding DNA (gift from Dr. Robert A. Weinberg, Massachusetts Institute of Technology).

**Isolation of neutrophils and NK cells.** Lung tissues were dissociated and digested with 1.5 mg ml$^{-1}$ collagenase (Sigma, St. Louis, MO) for 1 h at 37 °C. Red blood cells were then lysed by ACK lysis buffer (Gibco, Gaithersburg, MD). Neutrophils were purified using anti-Ly6G magnetic beads (Miltenyi Biotech, Auburn, CA) according to the manufacturer's instructions. NK cells were purified from spleen cells using NK cell isolation kit (Miltenyi Biotech, Auburn, CA) or from lung dissociated cells using the CD49b (DX5) MicroBeads (Miltenyi Biotech, Auburn, CA). The purities for isolated neutrophils and NK cells were above 95% and 90%, respectively, as analyzed by flow cytometry.

**Bioluminescence imaging.** Bioluminescence imaging (BLI) of the luciferase activity in tumor cells was used to monitor the tumor metastasis progression with a Xenogen IVIS system. Before performing the BLI, the mice were anesthetized under 2.5% isoflurane and administered with D-luciferin (150 mg kg$^{-1}$; Gold Biotechnology, St. Louis, MO) by intraperitoneal (i.p.) injection 10 min before the imaging. At the end time point, the lungs were harvested in a 24-well plate in 150 μg ml$^{-1}$ D-luciferin diluted in PBS for bioluminescence imaging. Image exposure times were between 10 s and 1 min, depending on the signal strength. Light emission from the region of interest was quantified as photons s$^{-1}$ cm$^{-2}$ steradian$^{-1}$.

**Neutrophil depletion and G-CSF treatment.** For neutrophil depletion studies, the mice were administrated with control IgG or anti-Ly6G (Clone 1A8; Bio X Cell, West Lebanon, NH) at a dose of 12.5 μg per mouse per day for 10 consecutive days.

For in vivo G-CSF administration, recombinant mouse G-CSF (Pepro tech, Rocky Hill, NJ) was i.p. injected daily at the dose of 2.5 μg per mouse.

**NK cell depletion.** To deplete NK cells in NOD-*scid* mice, anti-asialo GM1 (Biolegend #146002) was i.p. injected to the recipient mice at a dose of 25 μl per mouse every 3 days. To deplete NK cells in B6-*scid* mice, anti-NK1.1 antibody (Bio X cell # BE0036) was given to the recipient mice at a dose of 25 μg per mouse by i.p. injection every 3 days.

**ROS measurement.** Once the mice were euthanized, their lungs were dissected, put in a 24-well plate with 1 ml PBS and kept in an incubator for 10 mins at 37 °C. Then 2 μl L-012 (Sigma # SML2236, 5 mg ml$^{-1}$ in H$_2$O) was added to each well and the bioluminescence images were captured with a Xenogen IVIS system. Light emission from the region of interest was quantified as photons s$^{-1}$ cm$^{-2}$ steradian$^{-1}$.

**Flow cytometry.** For NK cell surface marker analysis, cells were suspended in staining buffer (PBS, 2% FBS) at a concentration of $<2 \times 10^7$ cells ml$^{-1}$ and 100 μl of suspension was incubated with fluorochrome-conjugated CD45, CD3e, CD49b, NKp46, Ly6G, and CD107a antibodies (Biolegend, San Diego, CA) for 30 min on ice. The dilution for antibodies used in flow cytometric analysis is 1:200. Cells were washed twice with staining buffer. Fluorescence intensity was measured by flow cytometry (FACSymphony A5, BD Immunocytometry, San Jose, CA). For detection of intracellular IFNγ, cells were primed with PMA (100 ng ml$^{-1}$) and ionomycin (1 μg ml$^{-1}$) for 4 h. Then the cells were fixed after surface staining, permeabilized with Cytofix/Cytoperm (BD Biosciences, San Jose, CA) and stained with an anti-IFNγ antibody (Biolegend, San Diego, CA). Data were analyzed using FlowJo software.

**NK cell and neutrophil co-culture.** NK cells ($4 \times 10^5$ cells) derived from spleens of NOD-*scid* mice were premixed with or without isolated neutrophils at a ratio of 1:1 for 2 h, and the cell mixture was then added to luciferase-labeled AT3 cells or stained YAC-1 cells at the NK cell: target ratios of 5:1, 10:1 or 20:1 for tumoricidal assay. To determine the mechanisms underlying neutrophil-mediated NK cell suppression, ROS inhibitors HDC (300 μg ml$^{-1}$), apocynin (100 μM) or catalase (1000 U ml$^{-1}$), or NO synthase inhibitor 1400 W dihydrochloride (10 μg ml$^{-1}$) was added when NK cells were premixed with neutrophils.

**Neutrophil conditioned medium.** Neutrophils were freshly isolated from lungs of E0771-*g-csf*-bearing NOD-*scid* mice, and were primed with PMA (100 ng ml$^{-1}$) for 1 h. Then $3 \times 10^6$ cells were re-suspended in 3 ml RPMI 1640 medium and incubated overnight. Finally, the supernatant was collected and mixed with an equal volume of RPMI 1640 complete medium as the neutrophil conditioned medium (CM). To determine the effects of neutrophil CM to kill tumor cells and suppress NK cells, 1 ml neutrophil CM was added to $3 \times 10^4$ AT3 cells or $6 \times 10^5$ NK cells for further tests.

**Tumoricidal assay.** For the cytotoxicity assay, purified NK cells, neutrophils or the premixed NK cells and neutrophils were added to luciferase-labeled AT3 cells ($3 \times 10^4$ cells) at the effector: target ratios of 5:1, 10:1 or 20:1. They were incubated in 1 ml complete RPMI 1640 medium in a 24-well-plate for 20 h. The luminescence was

then detected by luminometer in a SpectraMax® microplate reader (Molecular Devices, San Jose, CA). The tumor cell killing was calculated by the following formula: cell killing (%) = (1-[luminescence of target cells with effector cells]/ [luminescence of target cells only]) × 100%. Alternatively, YAC-1 cells ($1 \times 10^4$ cells) were first stained with CellTrace CFSE (Thermo Fisher Scientific, Waltham, MA), and were then incubated with effector cells at effector: target ratios of 1:1, 5:1, 10:1 or 20:1 in 1 ml complete RPMI 1640 medium in a 24-well-plate for 4 h. The percentages of YAC-1 cell killing were analyzed by flow cytometry. All data shown are the mean ± SD of four biologically independent cell cultures.

**Statistics and reproducibility.** The Prism software (version 7.04, GraphPad Software Inc.) was used for all statistical analyses. All data represent mean ± SEM or mean ± SD as indicated. The details of the statistical tests carried out are indicated in the respective figure legends. Values were compared using either an unpaired two-tailed *t*-test to compare two groups or ordinary one-way ANOVA with Tukey's multiple comparisons test to compare the variance in three or more groups with one independent factor (e.g. treatment group). We employed two-way ANOVA when there were effects of two factors (e.g. treatment and time) on a dependent variable. *P* value smaller than 0.05 was considered significant, whereas greater than 0.05 was assigned as not significant (ns). Exact *p* values are provided in each figure. Two (indicated in figure legends of Supplementary Figs. 3, 6, 7, 10, and 11) or three independent experiments (all other figures and supplementary figures) were performed and similar results were obtained.

**Reporting summary.** Further information on research design is available in the Nature Research Reporting Summary linked to this article.

## Data availability
The source data underlying Figs. 1b–d, f, g, 2b, d–g, i, j, 3c, e–i, 4b–d, f–h, j, k, 5b–f, 6b–e, g, i and Supplementary Figs. 1b, c, e, 2a, b, 4a–i, 5b–d, 6b, 8a, b, d–f, h–j, 9b, c, 10b, c, 11a–c are provided as a Source Data file. All the other data supporting the findings of this study are available within the article and its supplementary information files and from the corresponding author upon reasonable request. Source data are provided with this paper.

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

## Acknowledgements

We would like to thank Dr. Scott I. Abrams (Roswell Park Comprehensive Cancer Center) for providing the AT3 breast cancer cell line (originally generated in Dr. Abrams's lab), and thank Dr. Robert A. Weinberg (Whitehead Institute for Biomedical

Research) for providing the lentiviral vector expressing the mouse *G-csf*. This work was supported by grants from the National Institutes of Health (R00-CA188093 to G.R., R37-CA237307 to G.R., P30-CA034196 to Dr. Edison Liu. and R24-OD026440 and R01-AI132963 to L.S.), and a grant from the U.S. Department of Defense (W81XWH-18-1-0013 to G.R.). Q.L. is supported by the Pyewacket Fund at The Jackson Laboratory. We appreciate Dr. Kevin Seburn for his critical editing of the manuscript and also thank the assistance from The Jackson Laboratory Scientific Service.

## Author contributions

G.R., P.L. and M.L. conceived the project, designed the study and performed the data analysis. P.L. and M.L. did the mouse work and in vitro studies, performed the statistical analysis and generated the figures. P.L., J.S., L.H., Z.G., and Q.L. performed the mouse work and flow cytometry. L.S. provided critical assistance on experimental design. G.R., P.L., M.L. and L.S. interpreted the data and wrote the paper.

## Competing interests

The authors declare no competing interests.
