## [Peer Review File · Nature Communications]

Reviewers' comments:

Reviewer #1 (Remarks to the Author): Neutrophils and metastasis

Peishan et al., present a study where lung neutrophils activity towards metastatic cancer cells is analysed in the context of different immune suppressed mouse models. The metastatic model used here is the experimental metastasis where cancer cells are directly seeded in high amount in a specific moment in time into the lung via circulation. The authors show in vivo that the use of GCSF either exogenously administrated or overexpressed in the metastatic cells used, results in conflicting pro-tumourigenic or anti-tumourigenic outcome in presence or absence of NK cells respectively. Neutrophils depletion in the context of GCSF exogenous expression confirm that the opposing outcome is depended on neutrophils.

The model to explain this phenomenon proposed by the authors is that neutrophils have suppressive activity toward NK and killing activity toward cancer cells, but normally they mainly target NK cells, therefore NK cells can no longer stop cancer cells. However, in absence of NK cells, neutrophils would turn against cancer cells. The evidences provided to support this model are mainly based on ex vitro data, which do not necessarily reproduce the in vivo situation, and alone cannot be used to support the proposed mechanism.

The suppressive activity of neutrophils on NK cells was well described in syngeneic animals (Spiegel et al., 2016). This contribute to their pro-tumourigenic role. Another activity was described to be the direct selective proliferative boost of the highly metastatic cancer cell pool (Wculek et al., 2015). Conversely, in ex vivo conditions, the toxic activity of neutrophils when co-culture with cancer cells was long observed (Gerrard et al., 1981) and it also shown in human neutrophils isolated from healthy donors, however this activity appeared strongly reduced when neutrophils were derived from cancer patients (Yan et al, 2014).

Neutrophils have been previously described to have potential dual role and the understanding of this switch is now a key question, but I do not find the proposed model well supported by the data presented.

Key comments:

1. Why lung neutrophils are not acting as immunosuppressive cells or killer cells without GCSF stimulation (Figure 2d E0771+IgG vs E0771+AntiLy6g)?
2. GCSF is known to force mobilization of more immature neutrophils and GCSF induced human neutrophils show distinct functional features (Spiekermann et al., 1997). Indeed, the FACS analysis in Supp Figure 4b show a level of Ly6G expression strongly reduced in the sample with exogenous GCSF expression compared to normal lung neutrophils. What is the level of maturity of GCSF induced neutrophils?
3. The authors cannot exclude the possibility that neutrophils in NSG have different characteristics compared to fully immunocompetent mice, and that those are responsible for their anti-tumour activity. To test this the key experiment needs to be the use anti-NK1.1 depleting antibody to selectively remove NK cells from syngeneic B16 and NOD-scid mice to test if this is sufficient to induce anti-tumour activity in lung neutrophils.
4. The ability of neutrophils to suppress NK cells activity was previously reported (Spiegel et al., 2016), it is not surprising that this contribute to cancer cells growth in the lung. It is also not surprising that the ex vivo tests shown in Figure 3d and 4a confirm this observation. On the other hand the tumouricidal activity of Neutrophils alone is not very clear from Figure 4a. AT3 cells seems to be more sensitive compared to YAC-1 cells, which seems not to reach even 5% of cell killing even in a ration 1:20 with neutrophils for 20h. I supposed that ratio on the bases of the information that NK:neutrophils are at the ratio 1:1, and the NK:cancer cell ration is indicated, it will be the same for neutrophils alone, however an absolute number of cells need to be provided. If cancer cells are overwhelmed by NK and neutrophils they are unlikely to maintain healthy in culture. AT3 cells show about 10-20% killing with a ratio of 20:1 neutrophils and, as it appears that this can only be shown if the system is pushed, there is no evidence that this activity will be

actually displayed in vivo.

5. It is hard to imagine how neutrophils would selectively target those two cell types specifically in the tissue context using in both cases a ROS mechanism. Also why would the presence of NK cells stop the ROS to act on cancer cells? Moreover, a mechanism of toxicity based on ROS would not be specific for a cancer cells in the middle of normal tissue cells, which should equally, if not more, be killed. Is the in vitro killing activity not shown if a non-cancer cells is used in the co-culture? Finally, the inhibitors used in the in vitro assay Apocynin and Catalase, are also inhibiting intracellular ROS in neutrophils, which represent a signalling mechanism, so the ROS inhibitors could also influence other neutrophils behaviour and activation.

6. Finally, in the context of primary tumour, which is the requirement for spontaneous metastasis, there are convincing evidences that the neutrophils pool is perturbed systemically, it would be relevant to know how much the presence of a primary tumour would impact on neutrophils pro- and anti-tumorigenic activity in presence or absence of NK cells.

In conclusion, the data presented in this study, even if they describe a very intriguing phenomenon as starting point, which is worth investigating, are currently insufficient to support the proposed model.

Reviewer #2 (Remarks to the Author): Expert in metastasis

Manuscript by Peishan et al.

In this study, Peishan et al. seek to investigate the highly debated dual role of neutrophils in the context of metastasis inhibition or promotion. They show that GCSF expanded neutrophils inhibit metastasis in NK cell deficient mice as compared to NK containing mice. Neutrophil depletion with neutralizing anti-Ly6G antibody abrogated the pro or anti-metastatic effects. The authors suggest that in the future that the nature of the patient's immune status may guide GCSF therapy, and neutrophil targeting approaches. While this paper is on an interesting and timely topic, there are potential issues related to experimental design that are inconsistent with the interpretations drawn by the authors, further exacerbated by a general lack of mechanistic depth. Furthermore, a few published studies have somewhat dampened novelty of this work.

Comments:

1) The authors have narrowly focused on the role of GCSF on neutrophils in vivo. It may be important to also determine if GMCSF is impacting other immune cells. For example, a previous study has shown that it decreases IL-10 producing macrophages and increases IL-12 producing macrophages and IFN γ producing CD4+ and CD8+ T cells (Morris et al. Oncotarget 2015)

2) Ogura et al. (Cancer Immunol Res 2018) showed that in NK cell-depleted mice, neutrophils acquire a tumor-promoting phenotype, characterized by upregulation of VEGF-A expression, which promotes tumor growth and angiogenesis. The authors should discuss this study in the context of theirs.

3) The neutrophil phenotypes in NK containing and NK deficient mice are not sufficiently characterized. Are the NK cell inhibitory neutrophils N1, N2, SiglecFhigh, etc.

4) The authors state that neutrophils by virtue of producing ROS mediates NK cell suppression and tumor killing. This assertion is supported by invitro co-culture experiments utilizing ROS inhibitors (Apocynin and catalase). Whether, lung-infiltrating neutrophils utilize ROS pathway to selectively

inhibit NK cells and cancer cells in the metastatic niche has not been demonstrated.

5) The link between neutrophils and NK cells in the context of metastasis has been investigated, with demonstration that neutrophils suppress NK cells (Spiegel et al. Cancer Discovery 2016). This somewhat reduces novelty of the present study.

6) In Fig. 3a, details are missing. Did the lungs have different metastatic burden, was this analysis performed in metastasis free lungs.

7) To explain the contribution of neutrophils in mediating metastasis in the presence or absence of NK cells. the authors propose a model with the following elements: 1) NK cells possess a more robust killing potential compared to neutrophils, 2) in the absence of NK cells, neutrophils confer a higher tumor killing potential resulting in metastasis suppression and 3) alternatively, in the presence of NK cells, neutrophils suppress the more potent NK cells and therefore exhibit metastasis promoting phenotype. However, the experimental data does not provide any evidence to support these assertions. The net killing potential of neutrophils and NK cells is not quantitated. The authors should determine if NK cells influence neutrophil function.

8) The discussion needs modification. The authors need to reconcile with previous studies on neutrophils. NK cells and metastasis. There is also no discussion on how neutrophils suppress and NK cells.

Reviewer #3 (Remarks to the Author): Expert in NK cells and cancer

Review of: "Dual roles of neutrophils in metastatic colonization are governed by the host immune system integrity"

In the manuscript provided by Peishan Li and colleagues, the authors claim that the anti- or pro-tumoral functions of neutrophils are dictated by the presence or absence of NK cells in the tumor-bearing host. For this, they use exogenous G-CSF or tumor-induced G-CSF as a mean to expand the neutrophil pool in mice bearing the mammary gland adenocarcinoma E0771 model. To ensure that the observed tumor modulating effects of neutrophils are mediated by the host immune system - specifically NK cells - they perform these experiments side-by-side in NOD-scid mice (lacking T and B cells) and in NSG mice (lacking T, B and NK cells). Based on these in-vivo experiments, they conclude that G-CSF expanded neutrophils inhibit metastasis in the presence of NK cells in NOD-scid mice, whereas they promote it in the absence of NK cells in NSG mice. The authors perform one set of experiments ex-vivo to claim that neutrophils inhibit the cytotoxic activity of NK cells via ROS production. The latter is not new and has been shown elsewhere in studies involving neutrophils from cancer patients (Bruno et al., 2019) and in mouse experimental models (Stiff et al., 2019).

The claims made in this manuscript could be interesting; however a large part of the conclusions remains correlative, and there are a number of technical shortcomings in the study design. The study is indeed based on a modified experimental model that is not well characterized throughout the paper, and that does not seem to be fit to address the authors' questions. The study could strongly benefit from revision, mainly targeted at improving the experimental models and designs used, more accurately describing and interpreting the data, and making the writing of the conclusions more specific and less broad.

Below, I have detailed several major comments:

Results Section 2

- The composition of the immune infiltrate of the E0771 tumor model should be characterized (at

the primary tumor and metastatic sites), in the presence or absence of G-CSF.

The majority of the references in the result section refer to studies using the 4T1 breast cancer model, a TNBC model that spontaneously metastasizes to the lungs and that is neutrophil-rich (Mosely et al., 2016). In this model, neutrophils spontaneously infiltrate the primary tumor and pre-metastatic organs with different time kinetics and well before metastasis is established (Ouzounova et al., 2017). In the scope of this study, to address the involvement of the different immune players, the authors should characterize the immune composition of the E0771 ER+ tumor model, how it changes throughout tumor growth and metastatic progression, and how it is affected by administration or secretion of G-CSF.

- The impact of G-CSF on NK cells in the mice is not addressed. It is reported in the literature that G-CSF downregulates NK-cell mediated cytotoxicity (Su et al., 2011), and that anti-G-CSF therapy enhances NK cell numbers and cytolytic activity (Morris et al., 2015). The authors provide evidence that neutrophil depletion differentially abrogates the observed effects of G-CSF on metastasis in NSG and in NOD-scid mice; however, there is no evidence whether these results are linked to an increase or a decrease in NK-cell functionality in NOD-scid mice.

- The rationale for using a modified experimental model is not clear. The E0771 model metastasizes spontaneously to the lungs when engrafted orthotopically (Kanda et al., Cancer Res., 2009). The authors should then favor using the E0771 Luc and the E0771 Luc g-csf orthotopic models in NOD-scid and NSG mice to investigate the impact of neutrophilia on tumor growth and metastasis.

Using the current experimental design, it is not possible to differentiate an intra-tumoral or an exogenous source of G-CSF, since it overlooks the capacity of tumors to polarize neutrophils into different activation states.

- From a technical standpoint, the authors should provide details and corresponding data on how the pre-metastatic window is defined in their experimental model.

- Figure 2d: The results obtained when neutrophils are depleted suggest that the hypothesized inhibitory effect on NK cells is reversible. To specifically link this effect to the presence of NK cells, NK-depletion experiments in NOD-scid mice should be performed to see if the results are reversed.

Results Section 3

- The conclusions of the experiments presented in Figure 3 are correlative only. The observed reduction in NK cell cytotoxicity could be due to G-CSF and should be addressed as mentioned above (possibly via neutrophil depletion in this experimental setting). To get a better understanding of the hypothesized neutrophil – NK cell interaction, it will be worth investigating if the observed effect is restricted to the pre-metastatic niche or if the same is happening at the primary tumor site.

Results Section 4

The results presented in this section and accompanying figures are incomplete and cannot be fully interpreted in their current shape:

- Figure 4a and 4c: There is a big difference in the capacity of neutrophils to induce AT3 cell killing. 9%, 15% and 20% are reported in figure 4a column 3, figure 4a column 6 and figure 4c column 1 respectively. This raises concerns about the robustness of the assay and the interpretation of the data. The methods section should provide additional details on how the assays were performed.
- Figure 4b: Functional tests (cell killing assays) should be performed to confirm that ROS blockade reverses NK-cell inhibition

General comments:

- The title is misleading and does not reflect the results of this study. It indeed claims that the "immune host integrity" dictates neutrophil function in cancer. Whereas all the experiments are conducted in immunocompromised hosts lacking major immune populations with clear roles in immuno-oncology. Experimental models using syngeneic mouse models should be considered.

- The study could highly benefit from testing the hypothesis in a tumor model that is highly infiltrated by neutrophils, rather than inducing neutrophilia by using G-CSF or other cytokines.

Response to Reviewers' questions

Reviewer #1 (Remarks to the Author): Neutrophils and metastasis

Peishan et al., present a study where lung neutrophils activity towards metastatic cancer cells is analysed in the context of different immune suppressed mouse models. The metastatic model used here is the experimental metastasis where cancer cells are directly seeded in high amount in a specific moment in time into the lung via circulation. The authors show *in vivo* that the use of G-CSF either exogenously administrated or overexpressed in the metastatic cells used, results in conflicting pro-tumourigenic or anti-tumourigenic outcome in presence or absence of NK cells respectively. Neutrophils depletion in the context of G-CSF exogenous expression confirm that the opposing outcome is depended on neutrophils.

The model to explain this phenomenon proposed by the authors is that neutrophils have suppressive activity toward NK and killing activity toward cancer cells, but normally they mainly target NK cells, therefore NK cells can no longer stop cancer cells. However, in absence of NK cells, neutrophils would turn against cancer cells. The evidences provided to support this model are mainly based on *ex vitro* data, which do not necessarily reproduce the *in vivo* situation, and alone cannot be used to support the proposed mechanism.

We appreciate this reviewer for raising these critical points. Accordingly, we probed into the neutrophil-NK cell-tumor cell tri-cellular interaction mechanism by *in vivo* monitoring of the infused tumor cell fates at the very early stage (0-4~8 hrs) of metastatic colonization, a time window when NK cells and neutrophils kill most of the invading tumor cells.

1. In NK cell-deficient host mice (NSG), we compared the E0771 vs E0771-*g-csf* models which is to determine how G-CSF-induced neutrophilia affects tumor cells *in vivo*.

As shown in the new data **Figures 3d-e**, within the first 8 hrs upon tumor cell injection, the tumor cell colonization was significantly suppressed in the neutrophil^{high} hosts (E0771-*g-csf* model). This result suggested that neutrophils function to restrain early invaded tumor cells in the absence of NK cells *in vivo*.

2. In NK cell-competent host mice (NOD-*scid*), we again compared the E0771 vs E0771-*g-csf* models to evaluate the effects of G-CSF-induced neutrophilia on both NK cells and tumor cells *in vivo*.

- 1) As shown in our new data **Figures 4e-h**, a significant reduction of effector NK cells (IFN γ ⁺, CD107a⁺) was detected in neutrophil^{high} hosts (E0771-*g-csf* model), whereas such a reduction can be largely reversed by neutrophil depletion (anti-Ly6G). This indicated that neutrophils suppress effector NK cells *in vivo*.
- 2) As in our new results (**Figures 3b-c**), neutrophil^{high} host condition (E0771-*g-csf*) favored the early tumor cell colonization within the first 4-hr time window by BLI live imaging. This suggested that neutrophils exert a net tumor-promoting effect in the presence of NK cells *in vivo*.

These new *in vivo* results well corresponded to our previous *ex vitro* data shown in the original Figure 4a, which therefore highly strengthen our conclusion that the metastasis-modulating effects of neutrophils depend on the host NK cell status.

The suppressive activity of neutrophils on NK cells was well described in syngeneic animals (Spiegel et al., 2016). This contribute to their pro-tumourigenic role. Another activity was

described to be the direct selective proliferative boost of the highly metastatic cancer cell pool (Wculek et al., 2015). Conversely, in *ex vivo* conditions, the toxic activity of neutrophils when co-culture with cancer cells was long observed (Gerrard et al., 1981) and it also shown in human neutrophils isolated from healthy donors, however this activity appeared strongly reduced when neutrophils were derived from cancer patients (Yan et al, 2014). Neutrophils have been previously described to have potential dual role and the understanding of this switch is now a key question, but I do not find the proposed model well supported by the data presented.

We fully agree with this reviewer on the multifaceted functions of neutrophils in regulating both tumor cells and the host immunity. We therefore included more references in our revised “introduction” and “discussion” sections for a more comprehensive summary of the controversial roles of neutrophils in cancer progression. In these previous studies, either pro-tumoral or anti-tumoral effect of neutrophils, was always overemphasized in each individual work. Our work does not aim to deny any individual previous conclusions, but underline that neutrophils can be simultaneously pro-tumoral and anti-tumoral and their net tumor-modulatory effect shown under a certain biological condition is a compromise of their opposite effects. Such a net tumor-modulatory effect by neutrophils is largely dependent on the complexity of tissue microenvironment, particularly the host NK cell status.

Based on the constructive critiques from reviewers and the editor, we have substantially revised our manuscript in the following main aspects: 1) conducting the *in vivo* experiments to delineate the modulatory effects of neutrophils on NK cells and tumor cells (**Figure 3b-e** and **Figure 4e-h**); 2) comparing the phenotype and function of neutrophils in mice with different immune system integrity, as well as in naïve and inflammatory host conditions (**Figure 3f-g** and **Supplementary Figure 11a-c**); 3) performing *in vivo* NK cell depletion as an alternative to NSG mice in creating the NK cell-deficient host condition (**Figure 1f-g**); 4) immunostaining of neutrophils, NK cells and tumor cells in lung sections, which provides a rationale for the cell ratios used in our *ex vivo* co-culture assays (**Supplementary Figure 7a-b**); 5) characterizing the ROS-mediated effects by neutrophils in more detail *in vitro* and *in vivo* (**Figure 6** and **Supplementary Figure 9**); 6) validating our main conclusions in orthotopic breast tumor models which develop spontaneous lung metastasis (**Figure 2c-g**, **Figure 2j**, **Supplementary Figure 4f-i** and **Supplementary Figure 5a-d**).

Details are elaborated in the following point-to-point answers to individual questions, as well as in the revised manuscript. Again, we sincerely appreciate this reviewer to raise these exceptional questions which helped us to have fundamentally improved the quality of this manuscript. We are now confident that our current data will well support our proposed model.

Key comments:

1. Why lung neutrophils are not acting as immunosuppressive cells or killer cells without GCSF stimulation (Figure 2d E0771+IgG vs E0771+AntiLy6g)?

We thank this reviewer to raise this critical question. As validated in our new study (**Supplementary Figure 2c**), E0771 is a non-inflammatory model and there were far fewer number of lung-infiltrating neutrophils in E0771-bearing mice compared to the E0771-*g-csf* model. In this non-inflammatory condition, the lung-infiltrating neutrophils exert both NK cell suppressive and tumor-killing effects though both at relatively low levels, and a net tumor-modulatory effect of neutrophils in this condition could be minimal (see our proposed model in **Figure 3a**). Therefore, depletion of neutrophils in such a non-inflammatory host condition did not show a significant effect on both metastatic colonization (currently **Figure 2i**) and NK cell activity (**Figure 4g-h**).

2. G-CSF is known to force mobilization of more immature neutrophils and G-CSF induced human neutrophils show distinct functional features (Spiekermann et al., 1997). Indeed, the FACS analysis in Supp Figure 4b show a level of Ly6G expression strongly reduced in the sample with exogenous G-CSF expression compared to normal lung neutrophils. What is the level of maturity of G-CSF induced neutrophils?

This is an excellent question. According to this advice, we compared the main functions: 1) NK cell suppression, and 2) tumoricidal activity of neutrophils isolated from tumor-free mice, non-inflammatory (E0771-bearing) and neutrophil^{high} inflammatory tumor-bearing condition (E0771-g-csf). As shown in **Supplementary Figure 8i** and **Figure 3g**, all neutrophils were comparable with regard to both NK cell suppressive and tumoricidal capacities. Therefore, in spite of a difference in Ly6G expression, neutrophils are functionally (at least the main functions in this work) unaltered regardless of G-CSF stimulation or not.

3. The authors cannot exclude the possibility that neutrophils in NSG have different characteristics compared to fully immunocompetent mice, and that those are responsible for their anti-tumour activity. To test this the key experiment needs to be the use anti-NK1.1 depleting antibody to selectively remove NK cells from syngeneic B16 and NOD-scid mice to test if this is sufficient to induce anti-tumour activity in lung neutrophils.

We appreciate this reviewer for pointing out these major concerns. To answer these:

- 1) We performed new experiments comparing the neutrophils isolated from immunocompetent (C57BL/6J) and immunodeficient mice (NOD-scid and NSG). Neutrophils isolated from all three host mice were indistinguishable at both phenotypical and functional levels, including their tumoricidal effect (**Figure 3f**), NK cell suppressive effect (**Supplementary Figure 11a**), ROS producing levels (**Supplementary Figure 11b**) and expression of genes relevant to the main functions of neutrophils (**Supplementary Figure 11c**).
- 2) We followed the advice from this reviewer and performed NK cell depletion using anti-asialo GM1 (**Figure 1f** and **Supplementary Figure 1f**) or anti-NK1.1 (**Figure 1g** and **Supplementary Figure 1g**) to evaluate the NK cell-dependency of neutrophils in regulating metastatic colonization. The data clearly showed that G-CSF-induced neutrophilia was pro-metastatic in NK cell-competent mice, but anti-metastatic when NK cells were depleted. These new results are well consistent with the data in our original submission using NK cell-deficient NSG mice as shown in **Figure 1d**.

Thus, neutrophils are phenotypically and functional comparable in host mice with and without NK cells. NK cell depletion converted the pro-metastatic effect of G-CSF-induced neutrophilia to anti-metastatic.

4. The ability of neutrophils to suppress NK cells activity was previously reported (Spiegel et al., 2016), it is not surprising that this contribute to cancer cells growth in the lung. It is also not surprising that the *ex vivo* tests shown in Figure 3d and 4a confirm this observation. On the other hand the tumoricidal activity of Neutrophils alone is not very clear from Figure 4a. AT3 cells seems to be more sensitive compared to YAC-1 cells, which seems not to reach even 5% of cell killing even in a ration 1:20 with neutrophils for 20h. I supposed that ratio on the bases of the information that NK:neutrophils are at the ratio 1:1, and the NK:cancer cell ration is indicated, it will be the same for neutrophils alone, however an absolute number of cells need to be provided. If cancer cells are overwhelmed by NK and neutrophils they are unlikely to maintain healthy in culture. AT3 cells show about 10-20% killing with a ratio of 20:1 neutrophils

and, as it appears that this can only be shown if the system is pushed, there is no evidence that this activity will be actually displayed *in vivo*.

We thank this reviewer to raise these very important concerns. We answered this question in the following aspects:

- 1) As we mentioned above, we fully agree on the multifaceted functions of neutrophils in regulating both tumor cells and the host immunity. In the previous studies, a single effect of neutrophils, either pro-tumoral or anti-tumoral, was always overemphasized. Our work underlined that neutrophils can be simultaneously pro-tumoral and anti-tumoral and their net tumor-modulatory effect under a certain biological condition would be a compromise of their opposite effects. The net tumor-modulatory effect by neutrophils is dependent on the complexity of tissue microenvironment, particularly the host NK cell status.
- 2) We now provided a rationale for the cell ratios used in our *ex vivo* assays by performing the immunostaining of the lung sections in tumor-free, non-inflammatory (E0771) and inflammatory (E0771-*g-csf*) tumor-bearing conditions (**Supplementary Figure 7a-b**). As shown in **Supplementary Figure 7a**, the *in situ* ratio of neutrophils to NK cells in tumor-free or non-inflammatory hosts was about 1:1, whereas it was increased to about 5:1 under inflammatory (E0771-*g-csf*) host condition. At the early metastatic stage, we further detected the spatial distribution of tumor cells within the lung microenvironment by immunostaining. Apparently, the frequency of tumor cells was far lower than that of neutrophils (**Supplementary Figure 7b**). In addition to our own evidence, we also searched the literature for the ratios applied in previous reports^{1,2,3,4}, so we then adopted the 5~20:1 and 10~20:1 ratios for neutrophils: tumor cells and NK cells: tumor cells, respectively, in our *ex vivo* work.
- 3) As suggested, we have included the neutrophil: NK ratio, neutrophil: NK: tumor cell ratio and the absolute tumor cell numbers in the figure legends for our *ex vivo* studies. The detailed culturing information was also described in the “Material and Methods” section. For all the *ex vivo* work, we did not find over-confluence of any types of cells.
- 4) We have new data to support tumor cell killing by neutrophils *in vivo*. As shown in **Figure 3d-e**, in NK cell-deficient NSG mice, within the first 8 hrs upon tumor cell injection, the tumor cell colonization was significantly suppressed in the neutrophil^{high} hosts (E0771-*g-csf* model). This result suggested that neutrophils function to restrain early invaded tumor cells in the absence of NK cells *in vivo*.
5. It is hard to imagine how neutrophils would selectively target those two cell types specifically in the tissue context using in both cases a ROS mechanism. Also why would the presence of NK cells stop the ROS to act on cancer cells? Moreover, a mechanism of toxicity based on ROS would not be specific for a cancer cells in the middle of normal tissue cells, which should equally, if not more, be killed. Is the *in vitro* killing activity not shown if a non-cancer cells is used in the co-culture? Finally, the inhibitors used in the *in vitro* assay Apocynin and Catalase, are also inhibiting intracellular ROS in neutrophils, which represent a signalling mechanism, so the ROS inhibitors could also influence other neutrophils behaviour and activation.

We appreciate these great questions. To answer these questions:

- 1) We propose that neutrophils are non-selectively targeting both tumor cells and NK cells via ROS. Therefore neutrophils are simultaneously effective in killing tumor cells and suppressing NK cells under a specified condition. The ultimate tumor-regulatory outcome would be a compromise of various cell-cell interactions including neutrophils-NK cells, neutrophils-tumor cells and NK cells-tumor cells. In the revised manuscript we have clearly

stated this key concept. A modified diagram (**Figure 3a**) will also help to better understand the tri-cellular interactions under distinct host conditions.

- 2) We agree that ROS may have an intracellular impact inside the neutrophils. By performing a new experiment using the neutrophil-derived conditioned medium (CM), we found the CM was also effective in killing tumor cells and suppressing NK cells (**Supplementary Figure 9a-c**). As expected, stimulation of H₂O₂ decomposition by catalase, but not NADPH oxidase inhibitors, reversed the effect of the neutrophil CM. Therefore, our data indicated that neutrophil-derived ROS are able to exert extracellular effects on tumor cells and NK cells, although we did not preclude the possibility of ROS-mediated intracellular changes in neutrophils. We have discussed this in the revised manuscript.
- 3) We further conducted a series of experiments validating the *in vivo* roles of ROS in neutrophil-mediated NK cell suppression and tumoricidal effect. As shown in **Figure 6a-e**, G-CSF-driven host inflammatory condition caused an elevated ROS level in the lung along with reduced proportion of effector NK cells, which was substantially reversed by NADPH oxidase inhibition by histamine dihydrochloride (HDC). Moreover, administration of HDC significantly abrogated neutrophilia-induced pro-metastatic effect in NK cell-competent NOD-*scid* mice, and anti-metastatic effect in NK cell-deficient NSG mice *in vivo* (**Figure 6f-i**). All these new results further pinpointed the crucial role of ROS in mediating neutrophils to modulate both NK cells and tumor cells *in vivo*.

6. Finally, in the context of primary tumour, which is the requirement for spontaneous metastasis, there are convincing evidences that the neutrophils pool is perturbed systemically, it would be relevant to know how much the presence of a primary tumour would impact on neutrophils pro- and anti-tumorigenic activity in presence or absence of NK cells. In conclusion, the data presented in this study, even if they describe a very intriguing phenomenon as starting point, which is worth investigating, are currently insufficient to support the proposed model.

We thank this reviewer to raise this excellent question. To address this,

- 1) We have now included spontaneous lung metastasis models to further delineate how host NK cell presence determines the metastasis-modulatory effects of neutrophils. As in **Figure 2c-g** and **Supplementary Figure 4f-i**, we assessed how *g-csf* overexpression in two tumor models (E0771 and AT3) influenced their spontaneous lung metastases in NK cell-competent and –deficient host mice. The results were well consistent with our previous experimental metastasis model that G-CSF-induced neutrophilia was pro-metastatic in NK cell-competent NOD-*scid* mice, but anti-metastatic in NK cell-deficient NSG mice in both E0771 and AT3 models. Further, we adopted the spontaneous lung metastasis model with the 4T1 murine breast tumor cell line. As shown in **Figure 2j**, depletion of neutrophils by anti-Ly6G significantly reduced spontaneous lung metastasis in NK cell-competent BALB/c and NOD-*scid* mice, whereas conversely enhanced the lung metastasis in NK cell-deficient NSG mice. All these three spontaneous metastasis models unambiguously showed that the host NK cell presence or not is a decisive factor for neutrophil-mediated modulation of spontaneous metastasis.
- 2) In the modified experimental metastasis models that used in our original submission, we indeed orthotopically implanted control tumor cells or *g-csf*-overexpressed tumor cells to induce primary tumors, for example in **Figure 2a-b**. At the pre-metastatic stage, luciferase-labelled tumor cells were intravenously injected and the metastatic colonization was then compared between the non-inflammatory and inflammatory host conditions.

Reviewer #2 (Remarks to the Author): Expert in metastasis

Manuscript by Peishan et al.

In this study, Peishan et al. seek to investigate the highly debated dual role of neutrophils in the context of metastasis inhibition or promotion. They show that G-CSF expanded neutrophils inhibit metastasis in NK cell deficient mice as compared to NK containing mice. Neutrophil depletion with neutralizing anti-Ly6G antibody abrogated the pro or anti-metastatic effects. The authors suggest that in the future that the nature of the patient's immune status may guide G-CSF therapy, and neutrophil targeting approaches. While this paper is on an interesting and timely topic, there are potential issues related to experimental design that are inconsistent with the interpretations drawn by the authors, further exacerbated by a general lack of mechanistic depth. Furthermore, a few published studies have somewhat dampened novelty of this work.

Comments:

1) The authors have narrowly focused on the role of G-CSF on neutrophils *in vivo*. It may be important to also determine if GM-CSF is impacting other immune cells. For example, a previous study has shown that it decreases IL-10 producing macrophages and increases IL-12 producing macrophages and IFN γ producing CD4 $^+$ and CD8 $^+$ T cells (Morris et al. Oncotarget 2015)

We appreciate this reviewer to point out this important concern. Accordingly,

- a. We conducted a comprehensive comparison of the lung immune profiles between control and G-CSF injected mice or mice bearing *g-csf*-overexpressed tumor cells. The new data (**Supplementary Figure 1a-c** and **Supplementary Figure 2a-b**) showed that either G-CSF injection or *g-csf*-overexpression only remarkably expanded neutrophils, but not other myeloid lineage cells (dendritic cells and monocytes/macrophages) or adaptive immune cells (CD4 $^+$ T cells, CD8 $^+$ T cells and B cells).
- b. Further, our data in **Figure 2h-i** showed that anti-Ly6G-based neutrophil depletion largely abolished the pro-metastatic or anti-metastatic effects induced by tumor cell *g-csf*-overexpression verifying the specific role of G-CSF is induction of neutrophils.
- c. As requested by another reviewer, we also evaluated the direct impact of G-CSF on the NK cell functions and the new data in **Supplementary Figure 10a-c** showed that G-CSF did not influence the effector NK cell percentage and the tumoricidal activity of NK cells.

2) Ogura et al. Cancer Immunol Res 2018) showed that in NK cell-depleted mice, neutrophils acquire a tumor-promoting phenotype, characterized by upregulation of VEGF-A expression, which promotes tumor growth and angiogenesis. The authors should discuss this study in the context of theirs.

We thank this reviewer to point out this important reference. Accordingly, we performed NK cell depletion in NOD-*scid* (**Figure 1f** and **Supplementary Figure 1f**) and B6-*scid* mice (**Figure 1g** and **Supplementary Figure 1g**) to evaluate the NK cell-dependency of neutrophils in regulating metastatic colonization. The data clearly showed that G-CSF-induced neutrophilia was pro-metastatic in NK cell-competent mice, but anti-metastatic when NK cells were depleted. These new results are well consistent with the data in our original submission using NK cell-competent and NK cell-deficient mice as shown in **Figure 1a-d**.

We were quite intrigued by the discrepancy between Ogura et al. and ours. We reasoned that the discrepancy may be caused by the different tissue environments studied (primary tumor vs

lung metastatic site) and different tumor types used (fibrosarcoma and lung carcinoma vs breast carcinoma). We have discussed this reference in the revised “Discussion” section.

3) The neutrophil phenotypes in NK containing and NK deficient mice are not sufficiently characterized. Are the NK cell inhibitory neutrophils N1, N2, SiglecF^{high}, etc.

This is a great question and we accordingly performed new experiments to comprehensively compare neutrophils isolated from NK cell-competent mice (C57BL/6J and NOD-*scid*) and NK cell-deficient mice (NSG). As shown in **Figure 3f** and **Supplementary Figure 11a-c**, neutrophils isolated from all three host mice were indistinguishable at both phenotypical and functional levels, including their tumoricidal effect (**Figure 3f**), NK cell suppressive effect (**Supplementary Figure 11a**), ROS producing levels (**Supplementary Figure 11b**) and expression of genes relevant to the main functions of neutrophils (N1/N2, immunosuppression, activation and expansion, and ROS related molecules) (**Supplementary Figure 11c**).

4) The authors state that neutrophils by virtue of producing ROS mediates NK cell suppression and tumor killing. This assertion is supported by invitro co-culture experiments utilizing ROS inhibitors (Apocynin and catalase). Whether, lung-infiltrating neutrophils utilize ROS pathway to selectively inhibit NK cells and cancer cells in the metastatic niche has not been demonstrated.

We appreciate this reviewer to raise this critical question. Therefore we conducted a series of experiments validating the *in vivo* roles of ROS in neutrophil-mediated NK cell suppression and tumoricidal effect. As shown in **Figure 6a-e**, the G-CSF-driven host inflammatory condition led to an elevated ROS level in the lung along with reduced percentages of effector NK cells, which were largely reversed by a NADPH oxidase inhibitor, histamine dihydrochloride (HDC). Moreover, administration of HDC significantly abrogated neutrophilia-caused pro-metastatic effect in NK cell-competent NOD-*scid* mice, and anti-metastatic effect in NK cell-deficient NSG mice *in vivo* (**Figure 6f-i**). All these new results pinpointed the crucial roles of ROS in mediating lung neutrophils to modulate both NK cells and tumor cells *in vivo*.

5) The link between neutrophils and NK cells in the context of metastasis has been investigated, with demonstration that neutrophils suppress NK cells (Spiegel et al. Cancer Discovery 2016). This somewhat reduces novelty of the present study.

We well recognized the previously reported multifaceted functions of neutrophils in regulating both tumor cells and the host immunity. In these previous studies, a single effect of neutrophils, either pro-tumoral or anti-tumoral, was always overemphasized in each individual study. Our work underlines that neutrophils can be simultaneously pro-tumoral and anti-tumoral and their net tumor-modulatory effect shown under a certain biological condition is a compromise of these opposite effects. Such a net tumor-modulatory effect by neutrophils is largely dependent on the complexity of the tissue microenvironment, particularly the host NK cell status.

Further, in previous studies neutrophil-mediated immunosuppression was primarily concentrated on T cells which are repressed by a variety of neutrophil-derived immunoregulatory factors^{5, 6, 7, 8}. In contrast, fewer studies centered on NK cells, the essential anti-tumor host immune cells in the organ metastatic niche. Limited evidence, such as Spiegel et al. 2016⁹, indicated that organ-infiltrating neutrophils serve to suppress the ability of NK cells to undergo functional activation, however, a mechanistic understanding of neutrophil-mediated NK cell inhibition is lacking. In our work, we revealed that neutrophils suppress NK cell effector functions at the lung metastatic niche mainly through ROS *in vitro* and *in vivo*.

6) In Fig. 3a, details are missing. Did the lungs have different metastatic burden, was this analysis performed in metastasis free lungs.

The model depicted in **Figure 3a** is proposed for a stage that the metastatic tumor cells just colonize in the pre-metastatic lung immune microenvironment. Under four different host conditions (NK cell-competent vs NK-cell deficient; non-inflammatory vs neutrophil^{high}), although the same number of tumor cells are originally disseminated into the lung, distinct metastatic outcomes are carried out. In the revised **Figure 3a**, we made the diagram clearer which helps the readers to better understand the different host conditions-caused distinct metastatic outcomes.

Furthermore, we included the data (**Supplementary Figure 3**) showing the pre-metastatic stage determination for both AT3 and E0771 models that used throughout our study.

7) To explain the contribution of neutrophils in mediating metastasis in the presence or absence of NK cells, the authors propose a model with the following elements: 1) NK cells possess a more robust killing potential compared to neutrophils, 2) in the absence of NK cells, neutrophils confer a higher tumor killing potential resulting in metastasis suppression and 3) alternatively, in the presence of NK cells, neutrophils suppress the more potent NK cells and therefore exhibit metastasis promoting phenotype. However, the experimental data does not provide any evidence to support these assertions. The net killing potential of neutrophils and NK cells is not quantitated. The authors should determine if NK cells influence neutrophil function.

We appreciate this reviewer to raise these excellent questions. To address these,

- A. We indeed compared the tumoricidal capacities between NK cells and neutrophils. The effector cell: target cell titrating experiments for both NK cells (**Supplementary Figure 1e**) and neutrophils (**Figure 3f**) showed that NK cells are far more effective than neutrophils in killing tumor cells *ex vivo*.
- B. In **Figure 4i-k**, it was quantitatively shown that NK cells were more tumoricidal than neutrophils, but such a tumoricidal activity of NK cells was reduced when neutrophils were added in the culture. These data showed that neutrophils impaired the tumoricidal activity of NK cells at a quantitative level. In our revised figure legends for the *ex vivo* experiments, we added more details including the absolute cell numbers which helps the readers to better understand the tri-cellular interactions at the quantitative level.
- C. We further probed into the neutrophil-NK cell-tumor cell tri-cellular interaction mechanism by *in vivo* monitoring of the infused tumor cell fates at the very early stage (0-4~8 hrs) of metastatic colonization, a time window when NK cells and neutrophils kill most of the invading tumor cells.

C-1) In NK cell-deficient host mice (NSG), we compared the E0771 vs E0771-*g-csf* models which is to determine how G-CSF-induced neutrophilia affects tumor cells *in vivo*.

As shown in the new data **Figure 3d-e**, within the first 8 hrs upon tumor cell injection, the tumor cell colonization was significantly suppressed in the neutrophil^{high} hosts (E0771-*g-csf* model). This result suggested that neutrophils function to restrain early invaded tumor cells in the absence of NK cells *in vivo*.

C-2) In NK cell-competent host mice (NOD-*scid*), we again compared the E0771 vs E0771-*g-csf* models to evaluate the effects of G-CSF-induced neutrophilia on both NK cells and tumor cells *in vivo*.

As shown in our new data **Figures 4e-h**, a significant reduction of effector NK cells (IFN γ ⁺, CD107a⁺) was detected in neutrophil^{high} hosts (E0771-*g-csf* model), whereas such a reduction can be largely reversed by neutrophil depletion (anti-Ly6G). This result indicated that neutrophils suppress effector NK cells *in vivo*.

As in our new results (**Figures 3b-c**), neutrophil^{high} host condition (E0771-*g-csf*) favored the early tumor cell colonization within the first 4-hr time window by BLI live imaging. This result suggested neutrophils exert a net tumor-promoting effect in the presence of NK cells *in vivo*.

These new *in vivo* results well corresponded to our previous *ex vivo* data shown in the original Figure 4a (currently **Figure 4i-k**), which therefore highly strengthen our conclusion that the metastasis-modulating effects of neutrophils depend on the host NK cell status.

- D. We now provided a rationale for the cell ratios used in our *ex vivo* assays by performing the immunostaining of the lung sections in tumor-free, non-inflammatory (E0771) and inflammatory (E0771-*g-csf*) tumor-bearing conditions (**Supplementary Figure 7a-b**). As shown in **Supplementary Figure 7a**, the *in situ* ratio of neutrophils to NK cells in tumor-free or non-inflammatory hosts was about 1:1, whereas it was increased to about 5:1 under inflammatory (E0771-*g-csf*) host condition. At the early metastatic stage, we further detected the spatial distribution of tumor cells within the lung microenvironment by immunostaining. Apparently, the frequency of tumor cells was far lower than that of neutrophils (**Supplementary Figure 7b**). In addition to our own evidence, we also searched the literature for the ratios applied in previous reports^{1, 2, 3, 4}, so we then adopted the 5~20:1 and 10~20:1 ratios for neutrophils to tumor cells and NK cells to tumor cells, respectively, in our *ex vivo* work.
- E. As mentioned above, we compared neutrophils isolated from NK cell-competent (C57BL/6J and NOD-*scid*) and NK cell-deficient mice (NSG). As shown in **Figure 3f** and **Supplementary Figure 11a-c**, neutrophils isolated from all three host mice were comparable at both phenotypical and functional levels. These results suggested the presence or absence of host NK cells did not significantly influence neutrophils.

8) The discussion needs modification. The authors need to reconcile with previous studies on neutrophils, NK cells and metastasis. There is also no discussion on how neutrophils suppress and NK cells.

We thank this reviewer for this concern. According to the advice, we have made a substantial revision in the introduction, main text and discussion sections about the interactive relationships among neutrophils, NK cells and tumor cells in metastasis.

Reviewer #3 (Remarks to the Author): Expert in NK cells and cancer

Review of: "Dual roles of neutrophils in metastatic colonization are governed by the host immune system integrity"

In the manuscript provided by Peishan Li and colleagues, the authors claim that the anti- or pro-tumoral functions of neutrophils are dictated by the presence or absence of NK cells in the tumor-bearing host. For this, they use exogenous G-CSF or tumor-induced G-CSF as a mean to expand the neutrophil pool in mice bearing the mammary gland adenocarcinoma E0771 model. To ensure that the observed tumor modulating effects of neutrophils are mediated by the host immune system - specifically NK cells - they perform these experiments side-by-side in NOD-*scid* mice (lacking T and B cells) and in NSG mice (lacking T, B and NK cells). Based on these *in-vivo* experiments, they conclude that G-CSF expanded neutrophils inhibit metastasis in the presence of NK cells in NOD-*scid* mice, whereas they promote it in the absence of NK cells in NSG mice. The authors perform one set of experiments *ex-vivo* to claim that neutrophils inhibit the cytotoxic activity of NK cells via ROS production. The latter

is not new and has been shown elsewhere in studies involving neutrophils from cancer patients (Bruno et al., 2019) and in mouse experimental models (Stiff et al., 2019).

The claims made in this manuscript could be interesting; however a large part of the conclusions remains correlative, and there are a number of technical shortcomings in the study design. The study is indeed based on a modified experimental model that is not well characterized throughout the paper, and that does not seem to be fit to address the authors' questions. The study could strongly benefit from revision, mainly targeted at improving the experimental models and designs used, more accurately describing and interpreting the data, and making the writing of the conclusions more specific and less broad.

We appreciate this reviewer for her/his acknowledgment of the major conclusions that we made, as well as the critiques raised to help improve the quality of this work.

Based on the constructive comments from this reviewers, we have substantially revised our manuscript in the following main aspects: 1) conducting *in vivo* experiments to delineate the modulatory effects of neutrophils on NK cells and tumor cells; 2) comparing the phenotype and function of neutrophils in mice with different immune system integrity, as well as in non-inflammatory and inflammatory host conditions; 3) performing *in vivo* NK cell depletion as an alternative to NSG mice in creating the NK cell-deficient host condition; 4) performing immunostaining of neutrophils, NK cells and tumor cells in lung sections to provide a rationale for the cell ratios used in our *ex vivo* co-culture assays; 5) characterizing the ROS-mediated effects by neutrophils in more details *in vitro* and *in vivo*; 6) validating our main conclusions in orthotopic breast tumor models which develop spontaneous lung metastasis; 7) revising our title, introduction and discussion to make a more comprehensive summary of the research background and state our conclusions in a more accurate and specific manner.

We are confident that these changes have tremendously solidified our work and improved the quality of our manuscript.

Below, I have detailed several major comments:

Results Section 2

1. The composition of the immune infiltrate of the E0771 tumor model should be characterized (at the primary tumor and metastatic sites), in the presence or absence of G-CSF.

The majority of the references in the result section refer to studies using the 4T1 breast cancer model, a TNBC model that spontaneously metastasizes to the lungs and that is neutrophil-rich (Mosely et al., 2016). In this model, neutrophils spontaneously infiltrate the primary tumor and pre-metastatic organs with different time kinetics and well before metastasis is established (Ouzounova et al., 2017). In the scope of this study, to address the involvement of the different immune players, the authors should characterize the immune composition of the E0771 ER+ tumor model, how it changes throughout tumor growth and metastatic progression, and how it is affected by administration or secretion of G-CSF.

We appreciate this reviewer to raise this critical question. Accordingly,

1) We have performed a comprehensive analysis of the immune profiles in the E0771 model. First, by immunostaining of lung sections in **Supplementary Figure 2c**, it was shown that E0771 tumor alone did not induce a significant inflammatory response as reflected by

comparable numbers of lung-infiltrating neutrophils between tumor-free and E0771 tumor-bearing mouse lungs.

- 2) Further, by flow cytometry we compared the immune infiltrates in the lungs, primary tumors and peripheral blood in E0771 model with and without *g-csf*-overexpression. As shown in **Supplementary Figure 2a-b**, overexpression of *g-csf* in E0771 cells caused a striking increase of neutrophils in the lung, with only mild changes in other myeloid lineage cells (monocytes/macrophages and dendritic cells). At the primary tumor sites, a significantly higher number of dendritic cells were detected in E0771-*g-csf* tumors than that in the control E0771 tumors.
- 3) Moreover, exogenous injection of G-CSF similarly induced a striking induction of neutrophils, but not other myeloid cells or adaptive immune cells (CD4⁺ T cells, CD8⁺ T cells and B cells) (**Supplementary Figure 1a-c**).
- 4) Our previous data in **Figure 2d-e** (currently **Figure 2h-i**) showed that anti-Ly6G-based neutrophil depletion largely abolished the pro-metastatic or anti-metastatic effects induced by tumor cell *g-csf*-overexpression pinpointing the specific role of G-CSF is induction of neutrophils.
- 5) Above data suggested that E0771-*g-csf* model is similar to the well-studied 4T1 model^{9, 10} with regard to the dynamic of immune infiltrates, particularly in neutrophil profile, during metastatic progression. As suggested by another reviewer, we have also included the well-studied 4T1 spontaneous lung metastasis model (**Figure 2j**). The results were well consistent with the conclusions drawn from the E0771 model.

2. The impact of G-CSF on NK cells in the mice is not addressed. It is reported in the literature that G-CSF downregulates NK-cell mediated cytotoxicity (Su et al., 2011), and that anti-G-CSF therapy enhances NK cell numbers and cytolytic activity (Morris et al., 2015). The authors provide evidence that neutrophil depletion differentially abrogates the observed effects of G-CSF on metastasis in NSG and in NOD-scid mice; however, there is no evidence whether these results are linked to an increase or a decrease in NK-cell functionality in NOD-scid mice.

We thank this reviewer to raise these important concerns. As suggested,

- 1) We have conducted new *ex vivo* experiments to evaluate the direct effect of G-CSF on NK cell functionality. As shown in **Supplementary Figure 10a-c**, pre-treatment with recombinant mouse G-CSF did not significantly alter the effector NK cell percentage within the bulk NK cells, and the tumorcidal activity of NK cells. We discussed the literature in the “discussion” section.
 - 2) We accordingly characterized the effector NK cell changes when neutrophils were depleted. As shown in **Figure 4e-h**, anti-Ly6G-based neutrophil depletion largely reversed the significant reduction of effector NK cells (IFN γ ⁺, CD107a⁺) caused by neutrophil^{high} host condition (E0771-*g-csf* model). Therefore, neutrophil depletion indeed reversed tumor cell *g-csf* overexpression-induced NK cell suppression.
3. The rationale for using a modified experimental model is not clear. The E0771 model metastasizes spontaneously to the lungs when engrafted orthotopically (Kanda et al., Cancer Res., 2009). The authors should then favor using the E0771 Luc and the E0771 Luc G-CSF orthotopic models in NOD-scid and NSG mice to investigate the impact of neutrophilia on tumor growth and metastasis.

Using the current experimental design, it is not possible to differentiate an intra-tumoral or an

exogenous source of G-CSF, since it overlooks the capacity of tumors to polarize neutrophils into different activation states.

We appreciate all these excellent questions raised by this reviewer. To address them,

- 1) The rationale for using the modified experimental metastasis model, as depicted in the diagram in **Figure 2a**, is to determine how neutrophil^{high} inflammatory host condition differs from the non-inflammatory host condition in accommodating metastatic tumor cells, and their dependency on host NK cell presence. To this end, we performed intravenous injection of luciferase-labelled tumor cells at the pre-metastatic stage for the orthotopic tumors. That is to avoid any complications caused by the intrapulmonary tumor cells as mentioned by this reviewer.
- 2) We have now included the data (**Supplementary Figure 3**) showing the pre-metastatic stage determination for both AT3 and E0771 models that used throughout our study.
- 3) In addition to the experimental metastasis model, we have now included spontaneous lung metastasis models to further delineate how host NK cell status determines the metastasis-modulatory effects of neutrophils. As in **Figure 2c-g** and **Supplementary Figure 4f-i**, we assessed how *g-csf* overexpression in two tumor models (E0771 and AT3) influenced their spontaneous lung metastases in NK cell-competent and –deficient host mice. The results were well consistent with our previous experimental metastasis model that G-CSF-induced neutrophilia was pro-metastatic in NK cell-competent NOD-*scid* mice, but anti-metastatic in NK cell-deficient NSG mice in both E0771 and AT3 models.
- 4) Further, we adopted the spontaneous lung metastasis model with the 4T1 murine breast tumor cell line. As shown in **Figure 2j**, depletion of neutrophils by anti-Ly6G significantly reduced spontaneous lung metastasis in NK cell-competent BALB/c and NOD-*scid* mice, whereas conversely enhanced the lung metastasis in NK cell-deficient NSG mice. All these three spontaneous metastasis models unambiguously showed that the host NK cell presence or not is a decisive factor for neutrophil-mediated modulation of metastasis.
4. From a technical standpoint, the authors should provide details and corresponding data on how the pre-metastatic window is defined in their experimental model.

As mentioned above, we have now included the data in NSG mice (**Supplementary Figure 3**) as an example, to show the pre-metastatic stage determination for both AT3 and E0771 models.

5. Figure 2d: The results obtained when neutrophils are depleted suggest that the hypothesized inhibitory effect on NK cells is reversible. To specifically link this effect to the presence of NK cells, NK-depletion experiments in NOD-*scid* mice should be performed to see if the results are reversed.

This is an excellent question. We followed the advice from this reviewer and performed NK cell depletion using anti-asialo GM1 (**Figure 1f** and **Supplementary Figure 1f**) or anti-NK1.1 (**Figure 1g** and **Supplementary Figure 1g**) to evaluate how host NK cell status determines the roles of neutrophils in regulating metastatic colonization. The data clearly showed that G-CSF was pro-metastatic in NK cell-competent mice, but anti-metastatic when NK cells were depleted. These new results are well consistent with the data in our original submission using NK cell-deficient NSG mice as shown in **Figure 1a-d**.

Results Section 3

6. The conclusions of the experiments presented in Figure 3 are correlative only. The observed reduction in NK cell cytotoxicity could be due to G-CSF and should be addressed as mentioned

above (possibly via neutrophil depletion in this experimental setting). To get a better understanding of the hypothesized neutrophil – NK cell interaction, it will be worth investigating if the observed effect is restricted to the pre-metastatic niche or if the same is happening at the primary tumor site.

We appreciate this reviewer to raise this critical concern. As suggested, we performed neutrophil depletion by anti-Ly6G for the study shown in the original Figure 3b-c (currently **Figure 4e-h** and **Supplementary Figure 8f-h**).

As shown in **Figure 4e-h**, a significant reduction of effector NK cells (IFN γ^+ , CD107a $^+$) was detected in neutrophil^{high} hosts (E0771-G-CSF model), whereas such a reduction can be largely reversed by anti-Ly6G-based neutrophil depletion. This result indicated that neutrophils suppress effector NK cells *in vivo*.

As advised by this reviewer, we also examined the NK cell suppression at the primary tumor site. The new data in **Supplementary Figure 8f-h** revealed that tumor cell *g-csf* overexpression or neutrophil depletion did not influence the effector NK status within the primary tumor microenvironment. Therefore, the role of neutrophils in NK cell suppression is restricted in the pre-metastatic niche but not at the primary tumor site in our study.

Results Section 4

The results presented in this section and accompanying figures are incomplete and cannot be fully interpreted in their current shape:

7. Figure 4a and 4c: There is a big difference in the capacity of neutrophils to induce AT3 cell killing. 9%, 15% and 20% are reported in figure 4a column 3, figure 4a column 6 and figure 4c column 1 respectively. This raises concerns about the robustness of the assay and the interpretation of the data. The methods section should provide additional details on how the assays were performed.

We apologize for the confusion to this reviewer due to the insufficient information provided in our figure legends and “materials and methods”. In the original Figure 4a (currently **Figure 4j**) column 3 and column 6, the ratios of neutrophils to AT3 tumor cells are 10:1 and 20:1, and the percentages of AT3 killed are $9.37 \pm 2.24\%$ and $15.87 \pm 2.75\%$, respectively. In the original Figure 4c (currently **Figure 5e**) column 1, the ratio of neutrophils to AT3 tumor cells is 20:1, and the percentage of AT3 killed is $19.41 \pm 2.23\%$. In the revised figure legends and methods, we have provided the additional details including all cell-cell ratios and the absolute cell numbers.

Additionally, we also provided a rationale for the cell ratios used in our *ex vivo* assays by performing the immunostaining of the lung sections in tumor-free, non-inflammatory (E0771) and inflammatory (E0771-*g-csf*) tumor-bearing conditions (**Supplementary Figure 7a-b**). As shown in **Supplementary Figure 7a**, the *in situ* ratio of neutrophils to NK cells in tumor-free or non-inflammatory hosts was about 1:1, whereas it was increased to about 5:1 under inflammatory (E0771-*g-csf*) host condition. At the early metastatic stage, we further detected the spatial distribution of tumor cells within the lung microenvironment by immunostaining. Apparently, the frequency of tumor cells was far lower than that of neutrophils (**Supplementary Figure 7b**). In addition to our own evidence, we also searched the literature for the ratios applied in previous reports^{1, 2, 3, 4}, so we then adopted the 5~20:1 and 10~20:1 ratios for neutrophils to tumor cells and NK cells to tumor cells, respectively, in our *ex vivo* work.

8. Figure 4b: Functional tests (cell killing assays) should be performed to confirm that ROS blockade reverses NK-cell inhibition

We thank this reviewer to raise this critical question. As suggested, we performed NK cell tumoricidal assay when the ROS pathway was inhibited. As shown in **Figure 5c**, stimulation of H₂O₂ decomposition by catalase, or inhibition of NADPH oxidase by histamine dihydrochloride (HDC) or apocynin, were all able to reverse neutrophil-mediated suppression of NK cells' tumoricidal activity.

We also conducted a series of experiments validating the *in vivo* roles of ROS in neutrophil-mediated NK cell suppression and tumoricidal effect. As shown in **Figure 6a-e**, the G-CSF-driven host inflammatory condition led to an elevated ROS level in the lung along with reduced percentages of effector NK cells, which were substantially reversed by a NADPH oxidase inhibitor-HDC. Moreover, administration of HDC significantly abrogated neutrophilia-induced pro-metastatic effect in NK cell-competent NOD-*scid* mice, and anti-metastatic effect in NK cell-deficient NSG mice *in vivo* (**Figure 6f-i**). All these new results pinpointed the crucial roles of ROS in mediating lung neutrophils to modulate both NK cells and tumor cells *in vivo*.

General comments:

9. The title is misleading and does not reflect the results of this study. It indeed claims that the "immune host integrity" dictates neutrophil function in cancer. Whereas all the experiments are conducted in immunocompromised hosts lacking major immune populations with clear roles in immuno-oncology. Experimental models using syngeneic mouse models should be considered.

This is a great suggestion on our manuscript title and we have accordingly changed "host immune system integrity" to "host NK cell status" to better reflect our findings.

In terms of the syngeneic mouse models,

1) We have the experimental metastasis model in immunocompetent C57BL/6J mice in **Figure 1b**;

2) We also included the spontaneous lung metastasis model with 4T1 murine breast tumor cell line in syngeneic BALB/c mice (**Figure 2j**).

Both models indicated that neutrophils are pro-metastatic in NK cell-competent host mice.

10. The study could highly benefit from testing the hypothesis in a tumor model that is highly infiltrated by neutrophils, rather than inducing neutrophilia by using G-CSF or other cytokines.

We appreciate this reviewer for raising this critical concern. As mentioned above, we adopted the spontaneous lung metastasis model with the 4T1 murine breast tumor cell line. As shown in **Figure 2j**, depletion of neutrophils by anti-Ly6G significantly reduced spontaneous lung metastasis in NK cell-competent BALB/c and NOD-*scid* mice, whereas conversely enhanced the lung metastasis in NK cell-deficient NSG mice. The results unambiguously showed that the host NK cell presence or not is a decisive factor for neutrophil-mediated modulation of spontaneous metastasis.

Reference:

1. Michelet X, *et al.* Metabolic reprogramming of natural killer cells in obesity limits antitumor responses. *Nature immunology* **19**, 1330-1340 (2018).
2. Liu C, *et al.* Expansion of spleen myeloid suppressor cells represses NK cell cytotoxicity in tumor-bearing host. *Blood* **109**, 4336-4342 (2007).
3. Granot Z, Henke E, Comen EA, King TA, Norton L, Benezra R. Tumor entrained neutrophils inhibit seeding in the premetastatic lung. *Cancer cell* **20**, 300-314 (2011).
4. Gershkovitz M, *et al.* Microenvironmental Cues Determine Tumor Cell Susceptibility to Neutrophil Cytotoxicity. *Cancer research* **78**, 5050-5059 (2018).
5. Coffelt SB, *et al.* IL-17-producing gammadelta T cells and neutrophils conspire to promote breast cancer metastasis. *Nature* **522**, 345-348 (2015).
6. Fridlender ZG, *et al.* Polarization of tumor-associated neutrophil phenotype by TGF-beta: "N1" versus "N2" TAN. *Cancer cell* **16**, 183-194 (2009).
7. Movahedi K, *et al.* Identification of discrete tumor-induced myeloid-derived suppressor cell subpopulations with distinct T cell-suppressive activity. *Blood* **111**, 4233-4244 (2008).
8. Youn JI, Nagaraj S, Collazo M, Gabrilovich DI. Subsets of myeloid-derived suppressor cells in tumor-bearing mice. *Journal of immunology* **181**, 5791-5802 (2008).
9. Spiegel A, *et al.* Neutrophils Suppress Intraluminal NK Cell-Mediated Tumor Cell Clearance and Enhance Extravasation of Disseminated Carcinoma Cells. *Cancer discovery* **6**, 630-649 (2016).
10. Kowanetz M, *et al.* Granulocyte-colony stimulating factor promotes lung metastasis through mobilization of Ly6G+Ly6C+ granulocytes. *Proceedings of the National Academy of Sciences of the United States of America* **107**, 21248-21255 (2010).

REVIEWERS' COMMENTS:

Reviewer #1 (Remarks to the Author):

Peishan et al., have performed in great amount of experiments to address my previous comments. All my concerns have been addressed. The data are clearly showing a context (host) dependent effect on neutrophils which display an anti-metastatic function specifically in absence of NK cells (which are now also antibody depleted in immune-competent animals).

As one last thought: in my opinion the only question which remain, but that is outside the scope of this work, is around how resident lung cells (fibroblast, alveolar cells macrophages...) are not killed by the non-specific ROS toxic effect shown in Cancer cells and NK cells. Maybe specific tissue compensatory/protection effects are in place, a dedicated study on this would be interesting.

In conclusion, I found that the addition of the substantial amount of data have now generated a compelling manuscript and the paper is suitable for publication. This is a valuable work and the data presented will be of great interest to the cancer community.

Reviewer #2 (Remarks to the Author):

The authors have done a reasonable job in experimentally addressing the concerns and have also discussed other published studies in light of their findings. Particularly, inclusion of in vivo NK cell depletion experiments in immunocompetent mice, utility of orthotopic breast tumor models that develop spontaneous lung mets and deep characterization of neutrophils in mice with and without NK cells has significantly strengthened the revised manuscript.

Reviewer #4 (Remarks to the Author):

The paper is greatly improved and all comments of the reviewer were addressed in the revised version of the manuscript adding more mechanistic depth.

Response to referees:

Reviewer #1 (Remarks to the Author):

Peishan et al., have performed in great amount of experiments to address my previous comments. All my concerns have been addressed. The data are clearly showing a context (host) dependent effect on neutrophils which display an anti-metastatic function specifically in absence of NK cells (which are now also antibody depleted in immune-competent animals).

As one last though: in my opinion the only question which remain, but that is outside the scope of this work, is around how resident lung cells (fibroblast, alveolar cells macrophages...) are not killed by the non-specific ROS toxic effect shown in Cancer cells and NK cells. Maybe specific tissue compensatory/protection effects are in place, a dedicated study on this would be interesting.

In conclusion, I found that the addition of the substantial amount of data have now generated a compelling manuscript and the paper is suitable for publication. This is a valuable work and the data presented will be of great interest to the cancer community.

We appreciate this reviewer raising this excellent comment. We agree that it will be a fascinating and important research direction to further investigate the influence of neutrophil-derived ROS on different lung resident cell populations during breast cancer progression, as well as in other inflammatory diseases. The variation of distinct lung resident cell subsets in their sensitivity to ROS will be particularly interesting. The results will help us to develop a deeper and more precise understanding of the lung biology in steady and pathological conditions.

We have added a related discussion in the last paragraph of the "Discussion" section of the manuscript file, with tracked changes.